# Context-Informed Neural ODEs Unexpectedly Identify Broken Symmetries: Insights from the Poincaré–Hopf Theorem

**In Huh** [1 2]   **Changwook Jeong** [3 4]   **Muhammad Ashraful Alam** [1]

## Abstract

Out-Of-Domain (OOD) generalization is a significant challenge in learning dynamical systems, especially when they exhibit *bifurcation*, a sudden topological transition triggered by a model parameter crossing a critical threshold. A prevailing belief is that machine learning models, unless equipped with strong priors, struggle to generalize across bifurcations due to the abrupt changes in data characteristics. Contrary to this belief, we demonstrate that context-dependent Neural Ordinary Differential Equations (NODEs), *trained solely on localized, pre-bifurcation, symmetric data and without physics-based priors*, can still *identify post-bifurcation, symmetry-breaking behaviors, even in a zero-shot manner*. We interpret this capability to the model's implicit utilization of topological invariants, particularly the *Poincaré index*, and offer a formal explanation based on the *Poincaré–Hopf theorem*. We derive the conditions under which NODEs can recover—or erroneously hallucinate—broken symmetries without explicit training. Building on this insight, we showcase a topological regularizer inspired by the Poincaré–Hopf theorem and validate it empirically on phase transitions of systems described by the Landau–Khalatnikov equation.

## 1. Introduction

The laws of physical systems are frequently expressed by differential equations rooted in dynamical systems theory. Across all scientific disciplines, a key objective lies in constructing accurate mathematical models that seamlessly integrate observational data with physical principles. Properly modeled dynamical systems allow for the prediction of rare events or even unobserved phenomena, providing scientific intuition and opportunities for groundbreaking discoveries. Recently, learning dynamical systems directly from data has emerged as a promising alternative to traditional modeling approaches, offering a partial automation of scientific discovery (Brunton et al., 2016; Fotiadis et al., 2023; Huh et al., 2020; Kirchmeyer et al., 2022; Mouli et al., 2024; Nzoyem et al., 2025; Yin et al., 2021a;b). To serve as viable alternatives to physics-based models, these approaches must demonstrate robust forecasting capabilities beyond their training domain, making Out-Of-Domain (OOD) generalization a critical challenge (Göring et al., 2024).

Formally, a continuous dynamical system is represented by a phase space Ordinary Differential Equation (ODE):

$$\dot{\mathbf{x}}(t) = f(\mathbf{x}(t); \mu), \ \mathbf{x} \in \mathcal{M}, \ \mu \in \mathbb{R}^n, \ f : \mathcal{M} \times \mathbb{R}^n \to T\mathcal{M},$$

where $\mathbf{x}(t)$ is a phase space state at time $t$, $\mu$ is a $n$-dimensional model parameter vector, $f(\cdot; \cdot)$ is a vector field of the $\mu$-parameterized ODE, and $\mathcal{M}$ is a phase manifold. Here, the model parameters $\mu$ refer to factors that characterize the physical environment of the system, such as the mass of a pendulum[1]. Trajectories of ODEs are given by

$$\mathbf{x}(T; \mathbf{x}(0), \mu) = \mathbf{x}(0) + \int_0^T f(\mathbf{x}(t); \mu)\mathrm{d}t,$$

where $\mathbf{x}(0)$ is called the initial condition. The goal of learning dynamics is to use machine learning models like Recurrent Neural Networks (RNNs) (Brenner et al., 2022; 2025; Vlachas et al., 2018) or Neural ODEs (NODEs) (Chen et al., 2018; Rubanova et al., 2019) to accurately approximate the unknown vector field $f(\cdot; \cdot)$ based on the trajectory data. Clearly, for a given trajectory over $T$, there are two degrees of freedom: the initial condition $\mathbf{x}(0)$ and the model parameter $\mu$. Therefore, OOD challenges in the learning dynamics problem can be categorized into two types: generalizations to the unseen $\mathbf{x}(0)$ and to the unseen $\mu$ (Mouli et al., 2024).

---

[1]Elmore Family School of Electrical and Computer Engineering, Purdue University [2]CSE Team, Samsung Electronics [3]Graduate School of Semiconductor Materials and Devices Engineering, UNIST [4]Artificial Intelligence Graduate School, UNIST. Correspondence to: In Huh <ihuh@purdue.edu>, Changwook Jeong <changwook.jeong@unist.ac.kr>, Muhammad Ashraful Alam <alam@purdue.edu>.

*Proceedings of the 42nd International Conference on Machine Learning*, Vancouver, Canada. PMLR 267, 2025. Copyright 2025 by the author(s).

[1]Therefore, we will use the terms environment and parameter interchangeably, though the latter specifically referring to a numerical value that characterizes the environment.

The challenges of OOD learning dynamics become particularly prominent in the presence of *bifurcations*, where the system undergoes abrupt qualitative transitions in behavior as a certain parameter of $\mu$ crosses a critical threshold (Arnold et al., 2013; Strogatz, 2018). Among the various types of bifurcations, *spontaneous symmetry breaking*—where symmetric states lose stability and give way to asymmetric states—stand out for their profound implications across diverse fields, including the Higgs mechanism in particle physics (Bernstein, 1974), phase transitions in condensed matter field theory (Aranson & Kramer, 2002), switching operations in nanoelectronic (Alam & Zagni, 2024) and nanophotonic devices (Hamel et al., 2015), and even the dynamics of generative diffusion models (Raya & Ambrogioni, 2023). The (symmetry-breaking) bifurcations induce significant topological changes in the dynamical system due to the creation and annihilation of equilibrium states, thereby intertwining OOD challenges related to both initial conditions and parameters. Given these dramatic changes, purely data-driven models are widely believed to face challenges when extrapolating across bifurcation points, largely due to a lack of invariant features (Ye et al., 2021) between pre- and post-bifurcation trajectories. Hence, existing studies have emphasized integrating physics-informed priors (García Pérez et al., 2023; Ghadami & Epureanu, 2018; Kalia et al., 2021) or collecting cross-parameter datasets (Li & Yang, 2024) to improve the identifiability of bifurcations.

However, in this work, we make the surprising discovery that *context-dependent free-form NODEs, trained exclusively on highly localized pre-bifurcation data, without any physics-based priors, can unexpectedly detect the presence of symmetry-breaking bifurcations and recover the post-bifurcation behavior of hidden symmetries, in a zero-shot manner.* We demonstrate how purely data-driven NODE models can identify the bifurcation phenomena through the topological invariant known as the *Poincaré index* (Brasselet et al., 2009; Strogatz, 2018), revealing that the models implicitly learn and leverage it as a key invariant feature of the data. To support this observation, we formalize our explanation using the *Poincaré–Hopf theorem* (Hopf, 1927; Milnor & Weaver, 1997) and derive the conditions under which NODEs can either infer or, intriguingly, hallucinate broken symmetry without explicitly learning them. Based on this insight, we present a proof-of-concept study introducing a novel Poincaré–Hopf regularization, demonstrated through its application to the Landau–Khalatnikov (LK) theory (Landau & Khalatnikov, 1954) as an exemplar case.

## 2. Preliminary

**Context-dependent free-form NODEs.** Free-form NODEs refer to vanilla NODEs (Chen et al., 2018; Rubanova et al., 2019), where the vector field is a neural network parameterized by trainable weights $\theta$:

$$\dot{\mathbf{x}}(t) = f(\mathbf{x}(t); \theta), \; \theta \in \mathbb{R}^m. \quad (1)$$

This formulation describes a system operating within a single environment characterized by a fixed physical model parameter $\mu$, which is implicitly encoded in a single weight vector $\theta$. To generalize NODEs to represent multiple environments, $\theta$ should be expressed as dependent on $\mu$ such that $\dot{\mathbf{x}}(t) = f(\mathbf{x}(t); \theta(\mu))$. In practice, however, the exact values of $\mu$ are often unknown. In such cases, the NODEs must also seek a latent vector $\xi_e$ that captures the information of the specific physical model parameter $\mu = \mu_e$ corresponding to each given environment indicator $e$:

$$\dot{\mathbf{x}}(t) = f(\mathbf{x}(t); \theta_e = \theta(\xi_e)).$$

The environment-aware weight $\theta_e = \theta(\xi_e)$ can be implemented using the Feature-wise Linear Modulation (FiLM) layer (Perez et al., 2018) or hypernetwork structure (Ha et al., 2017). For a specific example, the Context-informed Dynamics Adaptation (CoDA) model (Kirchmeyer et al., 2022) employs a hypernetwork with low-rank decomposition, such that $\theta_e = \theta(\xi_e) = \theta_c + W\xi_e$. Here, $\theta_c$ is the centered weight that shared across all trajectories while $\xi_e$, referred to as the *context vector*, is specifically optimized for each set of trajectories associated with a distinct environment $e$. Typically, $\dim \xi_e \ll \dim \theta = m$ and it is often assumed $\dim \xi_e = \dim \mu = n$. The matrix $W$, which is also shared across all environments, maps the low-dimensional context vector $\xi_e$ to the $m$-dimensional weight space.

Then, this context-informed NODE is trained with some standard loss functions such as the Mean Squared Error (MSE) between the predicted and ground truth trajectories:

$$\mathcal{L}(\theta_e, D_e) = \sum_{j=1}^{T/\Delta t} \sum_{i=1}^{N} \left\| \mathbf{x}_e^i(t^j) - \tilde{\mathbf{x}}_e^i(t^j; \mathbf{x}_e^i(0), \theta_c + W\xi_e) \right\|_2^2,$$

$$\tilde{\mathbf{x}}(t^j; \mathbf{x}_e^i(0), \theta_c + W\xi_e) = \mathbf{x}_e^i(0) + \int_0^{t^j} f(\tilde{\mathbf{x}}(t); \theta_c + W\xi_e)\mathrm{d}t,$$

for an environment-specific dataset $D_e$ consists of $N$ different $(T/\Delta t)$-length discretized trajectories. Because there are multiple environments $\mathcal{E}_{\mathrm{tr}}$ sampled with different parameter values in the training dataset, the final objective is given by a summation over $\mathcal{E}_{\mathrm{tr}}$:

$$\min_{\theta_c, W, \{\xi_e\}_{e \in \mathcal{E}_{\mathrm{tr}}}} \sum_{e \in \mathcal{E}_{\mathrm{tr}}} \mathcal{L}(\theta_c + W\xi_e, D_e) + \mathcal{R}(W, \xi_e), \quad (2)$$

where $\mathcal{R}(\cdot)$ is a regularizer defined as $\mathcal{R}(W, \xi_e) = \lambda_\xi \|\xi_e\|_2^2 + \lambda_\Omega \sum_{i=1}^m \|W_{i,:}\|_2$, which induces sparsity on $W\xi_e$ and encourages the influence of $\theta_c$ to be sufficiently large across different environments (Kirchmeyer et al., 2022). $\lambda_\xi$ and $\lambda_\Omega$ are hyperparameters. Note that both the architecture and loss function of the model *do not include any specific physical priors*, aside from standard assumptions such as sparsity. After training, the model can simulate

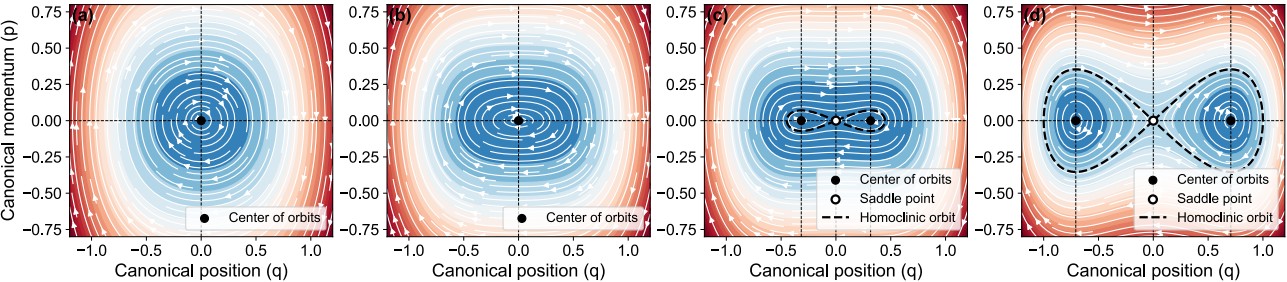

*Figure 1.* Phase portraits of the Hamiltonian system (3) for (a) $\mu = -0.5$, (b) $\mu = -0.1$, (c) $\mu = 0.1$, and (d) $\mu = 0.5$. Background contours represent the min-max normalized values of the Hamiltonian $\mathcal{H}(q, p; \mu)$ in (3). Red indicates higher relative values.

trajectories in a new environment either by adapting $\xi_e$ based on a small number of new observations (few-shot adaptation) or by directly modulating $\xi_e$ without further training (zero-shot exploration, analogous to latent space traversal in generative models (Song et al., 2023; Wei et al., 2024)).

**OOD in initial conditions.** The OOD condition for initial states is defined as a scenario where the model is trained on initial conditions $\mathbf{x}_e^i(0) \sim p_e^{\text{tr}}(\mathbf{x}(0))$, but the initial conditions of the test data fall outside the training support $\text{supp}(p_e^{\text{tr}})$ (Mouli et al., 2024). As highlighted in recent publications (Göring et al., 2024), OOD challenges in initial conditions become particularly significant when the dynamical system contains *separatrices*. These separatrices partition the phase space $\mathcal{M}$ into distinct regions, where trajectories originating from different domains exhibit qualitatively distinct behaviors, such as converging to different limit sets (e.g., in multistable systems) or transitioning between divergent motion patterns (e.g., from libration to rotation in pendulums). Consequently, each subdomain presents a challenging OOD scenario relative to the others. Formally, one can decompose such a system into $K$ disjoint subdomains $\mathcal{M}_i$ such that $\mathcal{M} = \bigcup_{i=1}^{K} \mathcal{M}_i \bigcup \partial\mathcal{M}$, where $\partial\mathcal{M}$ is the separatrix such that its Lebesgue measure is zero. Notably, if the entire training dataset is collected from one subdomain $\text{supp}(p_e^{\text{tr}}) \subseteq \mathcal{M}_k$ but the test data come from a different subdomain $\text{supp}(p_e^{\text{test}}) \subseteq \mathcal{M}_{i \neq k}$, this naturally creates the OOD condition, as the model will never encounter that particular regime during training, regardless of the sample size $N$ and time horizon $T$ of trajectories in the training dataset.

**OOD in model parameters.** The separatrix and multistable structures of phase spaces naturally define the boundaries of OOD problems in terms of initial conditions. Similarly, in the context of model parameters, *bifurcations* serve as key markers for delineating these boundaries in the parameter space. Formally, a parameter value $\mu_{\text{crit}}$ is called a bifurcation point of the family of parameterized ODEs $\dot{\mathbf{x}} = f(\mathbf{x}; \mu)$ if, for every neighborhood $U$ of $\mu_{\text{crit}}$ in the parameter space, there exist parameters $\mu_1, \mu_2 \in U$ such that $\dot{\mathbf{x}} = f(\mathbf{x}; \mu_1)$ and $\dot{\mathbf{x}} = f(\mathbf{x}; \mu_2)$ are *not* topologically (or qualitatively) equivalent in some neighborhood of the corresponding fixed points (or invariant set) in the phase

space. Consequently, the OOD condition for parameters arises when a model is trained on parameters $\mu_e^{\text{tr}} < \mu_{\text{crit}}$ but the support of the test data is $\mu_e^{\text{test}} > \mu_{\text{crit}}$[2].

Symmetry-breaking bifurcation, though purely arising from variations in parameters, inherently intertwines with OOD challenges in both model parameters and initial conditions. This complexity emerges because the broken symmetries introduce new asymmetric stable points with separatrices, complicating OOD generalization. Figure 1 shows a representative example of such a bifurcation, a Hamiltonian system with $\mathcal{H} = p^2/2 - \mu q^2/2 + q^4/4$, known as the double-well potential. The system's dynamics is given by

$$\dot{\mathbf{x}} = (\dot{q}, \dot{p}) = (\partial_p \mathcal{H}, -\partial_q \mathcal{H}) = (p, \mu q - q^3), \quad (3)$$

where $(q, p)$ represent the canonical coordinates, and $\mu$ is the model parameter. This system has a critical value of $\mu_{\text{crit}} = 0$: as shown in Figure 1 (a–b), when $\mu < 0$, the system exhibits a single family of orbits centered at the stable, symmetric center $(0, 0)$ (black dots). However, when $\mu > 0$, the bifurcation occurs as depicted in Figure 1 (c–d): the center of orbits at $(0, 0)$ becomes an unstable saddle point (white dots), and two new centers of orbits emerge at $(\pm\sqrt{\mu}, 0)$ (black dots), forming a double-well structure. Note that after this bifurcation, any infinitesimal oscillation around $(0, 0)$ will drive the system toward one of these newly emerged wells, thereby spontaneously breaking the original symmetry; although the system remains symmetric under $(q, p) \rightarrow (-q, -p)$, this *hidden* symmetry cannot be observed in practice. Furthermore, the homoclinic orbits (dashed lines) originating from the saddle point $(0, 0)$ delineate the separatrix, separating the left well, the right well, and the larger outer orbit structures. It naturally derives the OOD boundary under initial conditions.

## 3. Motivating Empirical Observations

### 3.1. Bifurcation Identification using NODEs

**Training procedure.** We trained the context-informed NODE (2) on the Hamiltonian system (3) under the *pre-bifurcation* regime exclusively. Specifically, we randomly

---

[2]For simplicity, this assumes a codimension-1 bifurcation with $n = 1$. For higher-dimensional cases, refer to Definition B.1.

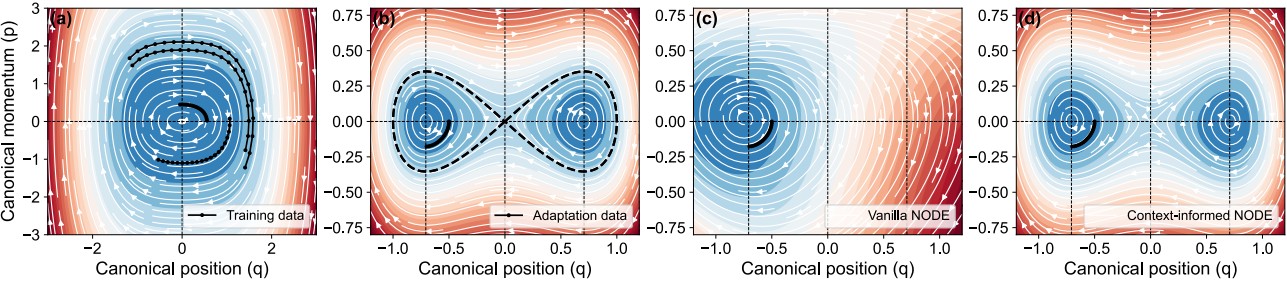

*Figure 2.* Examples of (a) training ($\mu = -0.5$) and (b) adaptation ($\mu = 0.5$, the left well case) trajectories. Predicted phase portraits of the (c) vanilla NODE and (d) context-informed NODE models, with background contours representing the numerically computed Hamiltonian profiles. In (c), the Hamiltonian is physically meaningless as the constructed dynamics is not a conservative system.

*Table 1.* MSEs ($\times 10^{-3}$) between the ground truth and models.

| SCENARIO | VANILLA | CONTEXT |
|---|---|---|
| LEFT WELL | 2427 $_{\pm 808.8}$ | 0.1517 $_{\pm 0.068}$ |
| RIGHT WELL | 1724 $_{\pm 365.1}$ | 0.2504 $_{\pm 0.033}$ |
| OUTER ORBIT | 3943 $_{\pm 904.8}$ | 0.1457 $_{\pm 0.065}$ |

sampled four initial conditions from the uniform distribution $(q(0), p(0)) \sim \mathcal{U}([-2.0, 2.0]^2)$ for each of the eight parameter values $\mu_e^{\text{tr}} \in \{-2.0, -1.75, -1.5, -1.25, -1.0, -0.75, -0.5, -0.25\}$. For each sampled initial condition, we simulated the dynamics with a time horizon $T = 2.0$ and a time step $\Delta t = 0.1$. This results in $|D_e| = 4$ trajectories per value of $\mu_e^{\text{tr}}$, yielding a total of $|D_e| \times 8 = 32$ training trajectories. It is important to note that all training data consists solely of single-orbit trajectories from the pre-bifurcation regime that exhibits the symmetric single-well structure. Figure 2 (a) shows the phase portraits of the training trajectories corresponding to $\mu = -0.5$ (refer to Appendix D for other cases and experimental details).

**One-shot adaptation.** After training the model with pre-bifurcation data, we adapted it using a single trajectory for $\mu = 0.5$, representing post-bifurcation data. We considered the following three *broken symmetry* scenarios, namely adaptations using: (i) a trajectory confined to the left well, (ii) one confined to the right well, and (iii) one outside the separatrix, traversing the outer orbit. These scenarios limit the model's exposure to the global structure of the phase space during adaptation, reflecting the symmetry breaking observed in real-world situations. For comparison, we also trained vanilla neural ODE models (1) under each scenario.

Figure 2 (b–d) compares the phase portraits of the ground truth (3), vanilla NODEs, and context-informed NODEs for the left well adaptation scenario near $(-\sqrt{\mu}, 0)$ (refer to Appendix D for other scenarios). The vanilla NODEs struggle to accurately capture the double-well structure, replicating the topology only within the region covered by the adaptation data. This result is consistent with recent theoretical findings suggested in (Göring et al., 2024). Note that it is a fundamental limitation of vanilla NODEs in OOD: increasing the number of post-bifurcation samples,

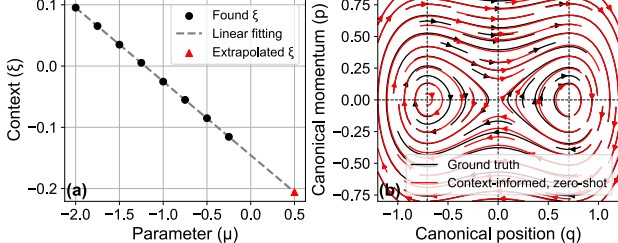

*Figure 3.* (a) Relationship between $\mu_e$ and $\xi_e$. (b) The phase portrait of the context-informed model ($\mu_e = 0.5$) extracted from (a).

yet remaining confined to the left well, does not improve the performance of this model. In contrast, the context-informed NODEs successfully reconstruct the entire phase topology across *all* scenarios. Remarkably, they successfully recover the hidden symmetry in the post-bifurcation regime, despite being trained exclusively on pre-bifurcation data and fine-tuned using only a single, chosen trajectory resulting from the broken symmetry. Table 1 shows the MSE between 32 test trajectories and NODE-predicted trajectories.

**Zero-shot exploration.** Figure 3 (a) illustrates the relationship between the ground truth parameters $\mu_e$ and the corresponding context $\xi_e$ constructed during the pre-bifurcation training. The plot reveals a nearly linear correlation between the two sets of values. Building on this observation, we extrapolated the context value for the post-bifurcation case where $\mu_e = 0.5$, obtaining $\xi_e = -0.2058$, and integrated it into the context-informed model. Note that this process was conducted without utilizing any post-bifurcation data adaptation, thus we refer to as zero-shot exploration. Figure 3 (b) presents the resultant phase portrait discovered in a zero-shot manner, showing it reproduces the ground truth one accurately (refer to Appendix D for different $\mu$ values).

**Automated generation of bifurcation diagrams.** Leveraging the zero-shot exploration capability and fully differentiable structure of NODEs, we can automatically construct bifurcation diagrams. To achieve this, we employ the neural-adjoint method (Ren et al., 2020) commonly used for inverse problems with neural networks. Specifically, for a given $\xi$, we identify fixed points $\mathbf{x}^*$ such that $f(\mathbf{x}^*) = \mathbf{0}$ by solving

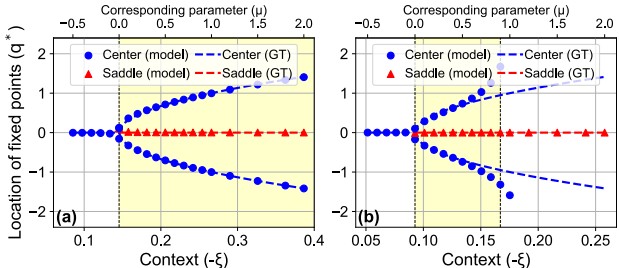

*Figure 4.* Bifurcation diagrams generated using context-informed models trained on (a) default and (b) localized domains.

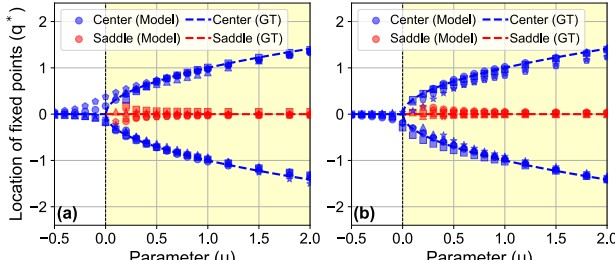

*Figure 5.* Bifurcation diagrams generated by context-informed models trained under two realistic settings: (a) the noisy setting, where small Gaussian noise ($\sigma = 0.02$) is added to the trajectories to mimic measurement noise. (b) the limited data setting, where only two initial conditions are used per parameter, compared to four in the default setting. Each setting is repeated five times with random initializations, indicated by distinct symbols.

the inverse problem $\mathbf{x}^* = \operatorname{argmin}_{\mathbf{x}} \| f(\mathbf{x}; \xi) - \mathbf{0} \|_2^2$ using the neural-adjoint method. Once the fixed points $\mathbf{x}^*$ are found, we compute the eigenvalues of the Jacobian matrix of $\mathcal{D}f(\mathbf{x}^*; \xi)$ at these points. The fixed points are then classified based on the characteristics of their eigenvalues, such as sign and the presence of real or imaginary components. By modulating $\xi$ and repeating this procedure, we construct a bifurcation diagram that intuitively illustrates the emergence or disappearance of fixed points. Refer to Appendix C for a detailed description of the proposed method. Figure 4 (a) presents the bifurcation diagram generated fully automatically, illustrating that a symmetry-breaking bifurcation occurs upon crossing the critical threshold, $\xi_{\mathrm{crit}}$.

**Robustness test.** To evaluate whether context-informed NODEs can identify the bifurcation structure under realistic conditions, including noisy observations and limited training data, we conducted additional robustness tests, as shown in Figure 5. These challenging settings revealed that the model remains capable of accurately identifying symmetry-breaking bifurcations, albeit with slightly increased variance compared to the ideal noise-free scenario.

**Context-informed NODEs identify broken symmetry.** The above presented experiments reveal that the context-informed model can infer the symmetry breaking behavior in the post-bifurcation regime, *despite being trained solely on pre-bifurcation data in a purely data-driven manner*. It showcases the model's capability to capture how parameter modulation influences the vector field, even without direct exposure to appropriate data. The most plausible explanation for this OOD learnability is that the context-aware model accurately captures the normal form of (3). Note that by using the Taylor expansion near $\xi = \xi_{\mathrm{crit}}$,

$$f(\mathbf{x}; \theta(\xi)) \simeq f(\mathbf{x}; \theta(\xi_{\mathrm{crit}})) + \nabla_\xi f(\mathbf{x}; \theta(\xi)) \big|_{\xi_{\mathrm{crit}}} \cdot (\xi - \xi_{\mathrm{crit}})$$
$$= f(\mathbf{x}; \theta(\xi_{\mathrm{crit}})) + \Phi(\mathbf{x}) \cdot (\xi - \xi_{\mathrm{crit}}),$$

where $\theta(\xi) = \theta_c + W\xi$ and $\Phi(\mathbf{x}) = \nabla_\xi f(\mathbf{x}; \theta_c + W\xi) \big|_{\xi = \xi_{\mathrm{crit}}}$ is the feature map. If the model is properly decomposed such that $f_0(\mathbf{x}) \simeq (p, -q^3)$ and $\Phi(\mathbf{x}) \simeq (0, \gamma q)$, where $\gamma$ is a constant, then the model can accurately replicate the bifurcation behavior. Indeed, when we compute

$\Phi(\mathbf{x})$, we find $\Phi(\mathbf{x}) \simeq (0, \gamma q)$, with $\gamma \simeq -8.3$ which corresponds closely to the slope shown in Figure 3 (a).

This analysis confirms that the model can approximate the parameter modulation as a feature map $\Phi(\cdot)$. However, to fully model the symmetry-breaking bifurcation, the model must accurately capture not only $\mu q$ but also $q^3$. This raises a pertinent question arises: Can the model accurately reconstruct the bifurcation of the dynamical system when trained on a highly constrained phase domain, such as the immediate vicinity of $(0,0)$? This presents a significant challenge, as it genuinely involves simultaneously addressing both parameter OOD and initial condition OOD issues.

### 3.2. Bifurcation Identification with Localized Domain

**Experimental setting.** We trained the context-informed NODE (2) on the Hamiltonian system (3) for pre-bifurcation scenarios, focusing on *localized* training domains. The experimental setup is consistent with the previous configuration, except that initial conditions were sampled from a narrower uniform distribution, $(q(0), p(0)) \sim \mathcal{U}([-0.5, 0.5]^2)$ (see Appendix E for details).

**Remarks on the experimental setting.** Within this restricted domain, the cubic term $q^3$ in (3) becomes less significant compared to the linear term $\mu q$, causing the training trajectories to resemble linear oscillators $(\dot{q}, \dot{p}) = (p, \mu q)$. This leads the model to predominantly observe linear behavior, potentially causing it to underestimate the role of the cubic term. Consequently, this limitation is expected to make the model prone to overlooking the formation of double-well structure caused by the influence of the cubic term. Note that, for the linear system $(\dot{q}, \dot{p}) = (p, \mu q)$, a bifurcation still occurs at $(0,0)$ when $\mu_{\mathrm{crit}} = 0$, transitioning from a center of orbits to a saddle point (refer to Remark B.2 of Appendix B for details). However, unlike (3), this system does not produce additional double centers of orbits, homoclinic loops, and outer orbit structure.

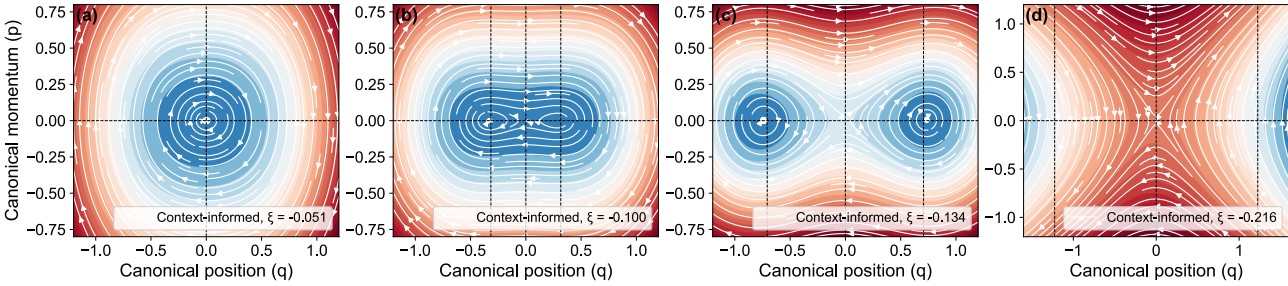

Figure 6. Phase portraits of the context-informed model trained with $(q(0), p(0)) \sim \mathcal{U}([-0.5, 0.5] \times [-0.5, 0.5])$, and constructed in a zero-shot manner by linearly extrapolating $\xi_e$, correspond to (a) $\mu_e = -0.5$, (b) $\mu_e = 0.1$, (c) $\mu_e = 0.5$, and (d) $\mu_e = 1.5$.

**Context-informed NODEs identify broken symmetry, unexpectedly.** We conducted the same zero-shot exploration procedure in Section 3.1. Figure 6 (b–d) presents the post-bifurcation phase portraits reconstructed in a zero-shot manner by inputting the explored context values $\xi_e$ for $\mu_e = 0.1, 0.5, 1.5$. We also plot the learned pre-bifurcation phase portrait at $\mu = -0.5$ as a reference in Figure 6 (a).

Remarkably, even though the model was trained only on localized linear-like data, it still reconstructs the broken symmetry, even for $\mu_e = 0.5$, where the double centers are located at $\pm\sqrt{0.5} \simeq 0.707 > 0.5$ (Figure 6 (c)). As noted earlier, it is not surprising that the model identifies the center-to-saddle transition, because a similar bifurcation is present in the linear system. If the model learns the local behavior near $(0, 0)$ with respect to parameter modulation, it can naturally capture this transition. What is truly notable, however, is that the model accurately predicts the emergence of the double-well structure along with the homoclinic loop.

Another noteworthy observation is that, as $\mu_e$ increases (i.e., as $-\xi_e$ increases), the model's double-well topology collapses, reverting to a single linear saddle point (see Figure 6 (d), and bifurcation diagram Figure 4 (b)). Note that the model from Section 3.1 retains the double-well structure throughout the range of $\mu > 0$, as shown in Figure 4 (a).

Additionally, we found that the context-informed model, *even when trained on a linear system $(\dot{q}, \dot{p}) = (p, \mu q)$ without the $q^3$ term, surprisingly produces a spurious double-well structure as well.* A detailed presentation and discussion of this finding are provided in Section 4.3. These findings suggest the presence of an intrinsic mechanism that drives the model to spontaneously generate a double-well structure, rather than correctly learning the functional form.

# 4. Insights from the Poincaré–Hopf Theorem

## 4.1. Poincaré–Hopf Index Theory

In this section and Appendix A, we briefly introduce the *Poincaré index*, an integer that can characterize topology of a vector field not only locally, but also globally, depending on the choice of a test contour (Strogatz, 2018). It helps answer questions such as: what types of fixed points can merge

during bifurcations? We then interpret the observations in Section 3.2 through the lens of the Poincaré index.

**Definition 4.1.** *(Poincaré index) Let $\mathcal{M}$ be an oriented smooth $d$-manifold and let $f : \mathcal{M} \to T\mathcal{M}$ be a smooth vector field. Suppose $\mathbf{x}^* \in \mathcal{M}$ is an isolated zero of $f$, i.e., $f(\mathbf{x}^*) = 0$ and there is a neighborhood $U$ of $\mathbf{x}^*$ with $f(\mathbf{x}) \neq 0$ for all $\mathbf{x} \in U \setminus \{\mathbf{x}^*\}$. Choose an oriented coordinate chart around $\mathbf{x}^*$ and a closed $d$-ball $D \subset \mathcal{M}$ centered at $\mathbf{x}^*$ such that $\mathbf{x}^*$ is the only zero of $f$ in $D$ and $f(\mathbf{x}) \neq 0$ for all $\mathbf{x} \in \partial D$ on the boundary $\partial D$. The Poincaré index of $f$ at $\mathbf{x}^*$, $\mathrm{Ind}(f, \mathbf{x}^*)$, is defined as the topological degree of the map $\Phi : \partial D \to \mathbb{S}^{d-1}$, $\Phi(\mathbf{x}) = f(\mathbf{x})/\|f(\mathbf{x})\|$:*

$$\mathrm{Ind}(f, \mathbf{x}^*) := \deg(\Phi) \in \mathbb{Z}.$$

*This integer is independent of the choice of chart and $D$.*

**Remark 4.1.** *For $\mathcal{M} = \mathbb{R}^2$, the Poincaré index of a zero $\mathbf{x}^*$ coincides with the winding number of the vector field $f$ around $\mathbf{x}^*$. Intuitively, this measures how many times and in which direction the vector field rotates around $\mathbf{x}^*$ as one traverses a simple closed test contour $\partial D$ that encircles $\mathbf{x}^*$ counterclockwise. For $f(\mathbf{x}) = (f_q(\mathbf{x}), f_p(\mathbf{x}))$, it is equal to*

$$\mathrm{Ind}(f, \mathbf{x}^*) = \frac{1}{2\pi} \oint_{\partial D} d\theta = \frac{1}{2\pi} \oint_{\partial D} \frac{-f_p df_q + f_q df_p}{f_q^2 + f_p^2}.$$

**Remark 4.2.** *For $\mathcal{M} = \mathbb{R}^2$, an isolated sink, source, center of orbits, and spiral each have a Poincaré index of $+1$, whereas a saddle point has a Poincaré index of $-1$.*

**Remark 4.3.** *For $\mathcal{M} = \mathbb{R}^2$, if a test contour $\Gamma$ contains multiple isolated zeros $\{\mathbf{x}_1^*, \mathbf{x}_2^*, \ldots, \mathbf{x}_k^*\}$, the index of $f$ along $\Gamma$ is $\mathrm{Ind}(f, \Gamma) = \sum_{i=1}^{k} \mathrm{Ind}(f, \mathbf{x}_i^*)$. This follows because $\Gamma$ can be continuously deformed into a new closed curve $\Gamma'$ that consists of $k$ small closed loops $\{\partial D_1, \partial D_2, \ldots, \partial D_k\}$, each surrounding one of the zeros $\{\mathbf{x}_1^*, \mathbf{x}_2^*, \ldots, \mathbf{x}_k^*\}$, connected by two-way bridges (see Figure 7 (a)). Considering that the contributions from the bridges cancel each other out, $\mathrm{Ind}(f, \Gamma) = \sum_{i=1}^{k} \mathrm{Ind}(f, \mathbf{x}_i^*)$ holds.*

**Theorem 4.1.** *(Poincaré–Hopf Theorem) Let $\mathcal{M}$ be a compact, oriented, smooth manifold without boundary, and let $f : \mathcal{M} \to T\mathcal{M}$ be a smooth vector field on $\mathcal{M}$ with finitely many isolated zeros $\{\mathbf{x}_1^*, \mathbf{x}_2^*, \ldots, \mathbf{x}_k^*\}$. Then, the sum of the Poincaré indices of $f$ at these zeros is equal to the Euler characteristic $\chi(\mathcal{M})$ of $\mathcal{M}$: $\sum_{i=1}^{k} \mathrm{Ind}(f, \mathbf{x}_i^*) = \chi(\mathcal{M})$.*

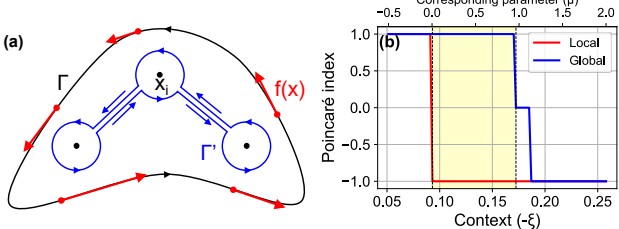

*Figure 7.* (a) Poincaré index of a closed orbit $\Gamma$ contains multiple fixed points. (b) Computed local ($r = 0.1$) and global ($r = 10$) Poincaré indices for the model in Section 3.2 (see Figure 4 (b)), where $r$ is the radius of a test contour centered at $(0, 0)$.

A formal proof can be found in textbooks on differential topology, such as (Milnor & Weaver, 1997). In addition, although Theorem 4.1 is presented in a generalized form, the following 2-dimensional statement is more directly applicable to interpret the situation observed in Section 3.2.

**Corollary 4.1.** *(Poincaré–Hopf for Closed Orbits) Let $f : \mathbb{R}^2 \to \mathbb{R}^2$ be a smooth vector field and let $\Gamma$ be a simple (non-self-intersecting) closed orbit of the dynamical system $\dot{\mathbf{x}} = f(\mathbf{x})$. Suppose that all fixed points (all zeros of $f$) inside $\Gamma$ are isolated. Then, the sum of the Poincaré indices of all fixed points inside $\Gamma$ is equal to $+1$: $\sum_{\mathbf{x}^* \in \text{int}(\Gamma)} \text{Ind}(f, \mathbf{x}^*) = +1$.*

*Proof.* A detailed proof can be found in standard textbooks, such as (Strogatz, 2018). Intuitively, because $\Gamma$ is an actual trajectory of the dynamical system $\dot{\mathbf{x}} = f(\mathbf{x})$, $f(\mathbf{x})$ is tangential at every $\mathbf{x} \in \Gamma$ (see Figure 7 (a)). It ensures $\text{Ind}(f, \Gamma)$ is equal to $+1$. Then, from Remark 4.3, $\text{Ind}(f, \Gamma) = \sum_{i=1}^{k} \text{Ind}(f, \mathbf{x}_i^*) = +1$, where $\mathbf{x}_1^*, \mathbf{x}_2^*, \ldots, \mathbf{x}_k^*$ are fixed points lying in the interior of $\Gamma$. □

### 4.2. Poincaré–Hopf as an Implicit Regularization

Note that any NODE represents a smooth vector field when using smooth activations such as tanh or swish. Then, assume that the context-informed model effectively captures the transition where the center of single orbits at $(0, 0)$ transforms into a saddle point. In this case, the emergence of the saddle point implies a change in the local Poincaré index from $+1$ to $-1$, according to Remark 4.2.

The context-informed model learns a foliation of closed orbits for $\xi < \xi_{\text{crit}}$ from the single-well training data (Figure 6 (a)). Now, assume that the model preserves at least one orbit $\Gamma$ encircling $(0, 0)$, under small perturbations in $\xi_{\text{crit}}$[3]. Then, the summation of the Poincaré indices within $\Gamma$ must be $+1$ by Corollary 4.1. Thus, to preserve the total index of $+1$ inside $\Gamma$, the model must generate additional fixed points whose combined Poincaré index sums to $+2$, offsetting the

---

[3]Empirically, we observed that this assumption often holds. It can be partially justified by noting that a small perturbation does not completely collapse the foliated orbits of conservative systems.

$-1$ from the newly formed saddle. These additional fixed points serve as centers of two orbits (see Figure 6 (b–c)).

However, if the outer orbit containing fixed points collapses due to a strong $\xi$ perturbation, the constraints from Corollary 4.1 no longer apply, and there is no need to maintain these additional centers (Figure 6 (d)). We computed the Poincaré index as a function of $\xi$ in Figure 7 (b), which predicts the lifetime of the double-well observed in Figure 4 (b) (highlighted in yellow for both plots).

Based on these observations, in Proposition 4.1, we formally derive the condition under which the context-informed NODE exhibits the symmetry-breaking bifurcation.

**Proposition 4.1.** *Let $f : \mathbb{R}^2 \times \mathbb{R} \to \mathbb{R}^2$ be a Hamiltonian vector field of the form $f(q, p) = (p, \mu q + \mathcal{P}(q))$, where $\mathcal{P}(q)$ is a smooth function such that $\mathcal{P}(0) = \mathcal{P}'(0) = 0$. Consider another smooth vector field $g : \mathbb{R}^2 \times \mathbb{R} \to \mathbb{R}^2$ for which there exists a smooth bijective map $\phi : \mathbb{R} \to \mathbb{R}$ such that $f$ and $g$ are $\delta$-close in the $\mathcal{C}^1$ sense:*

$$\sup_{\mathbf{x} \in U} \|g(\mathbf{x}; \phi(\mu)) - f(\mathbf{x}; \mu)\|_{\mathcal{C}^1(U)} \leq \delta,$$

*in some open interval $\mu \in (-\epsilon, \epsilon)$, where $U \in \mathbb{R}^2$ is some neighborhood of $(0, 0)$. Suppose that the dynamical system $\dot{\mathbf{x}} = g(\mathbf{x}, \xi)$ admits at least one closed orbit $\Gamma$ that encloses the isolated fixed point $\mathbf{x}^* \simeq (0, 0)$ of $g$, for some neighborhood of $\xi_{\text{crit}} = \phi(\mu_{\text{crit}})$. Then, $g$ undergoes the (generalized) symmetry breaking, at least locally near $\xi_{\text{crit}}$.*

See Appendix B for an informal proof. Proposition 4.1 points to an intriguing perspective: it only requires that the ground truth $f(\cdot; \mu)$ undergoes a center-to-saddle bifurcation, not necessarily the symmetry-breaking one. Such a transition could, for instance, result from a simple linear system as discussed. However, once the context-informed model $f(\cdot; \xi)$ learns this local behavior and exhibits an orbit encompassing this local region, the Poincaré–Hopf theorem compels the model to produce the additional fixed points.

### 4.3. Hallucinated Broken Symmetry

Inspired from Proposition 4.1, we present an experiment where the context-informed NODE identifies incorrect bifurcation behavior. In this experiment, we consider learning a linear system defined as $(\dot{q}, \dot{p}) = (p, \mu q)$. As briefly explained in Section 3.2, this system has a simple orbit structure for $\mu < 0$ and exhibits a center-to-saddle bifurcation at $(0, 0)$ when $\mu > \mu_{\text{crit}} = 0$, but does not produce the double-well structure. We simulated this linear system over $T = 2.0$ with $\Delta t = 0.1$, using four sampled initial conditions from $\mathcal{U}([-0.3, 0.3]^2)$, for each $\mu_e^{\text{tr}} \in \{-0.35, -0.25, -0.15, -0.05, 0.05, 0.15, 0.25, 0.35\}$, as training data (see Appendix F for details). Note that, in this setting, the model explicitly learns from data in the post-bifurcation regime, in contrast to the previous experiment.

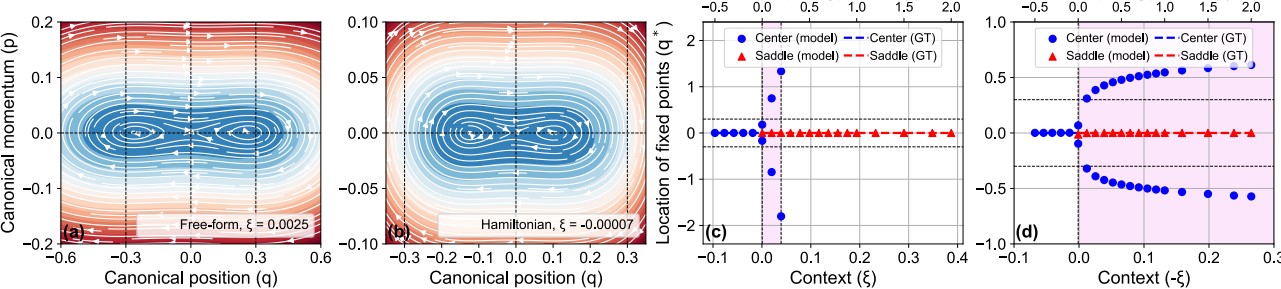

*Figure 8.* Phase portraits of (a) free-form and (b) Hamiltonian models trained with $(\dot{q}, \dot{p}) = (p, \mu q)$, and constructed in a zero-shot manner by linearly extrapolating $\xi_e$, correspond to $\mu_e = 0.01$. Bifurcation diagrams of (c) free-form and (d) Hamiltonian models.

We evaluated two different architectures of context-informed NODEs: the default free-form and its Hamiltonian version (Greydanus et al., 2019). The latter is defined as

$$\dot{\mathbf{x}} = (\dot{q}, \dot{p}) = f(\mathbf{x}, \theta_e) = (\partial_p \mathcal{H}(\mathbf{x}; \theta_e), -\partial_q \mathcal{H}(\mathbf{x}; \theta_e)),$$

where $\mathcal{H}(\mathbf{x}; \theta_e) = \mathcal{H}(\mathbf{x}; \theta_c + W\xi_e)$ is a context-informed learnable Hamiltonian (see Appendix F for details). Since $(\dot{q}, \dot{p}) = (p, \mu q)$ is also a Hamiltonian system, applying the Hamiltonian bias appears fundamentally appropriate.

Figure 8 (a–b) illustrates the phase portraits explored by context-informed models trained on the linear system. As shown, *despite being trained with both pre- and post-bifurcation data, the free-form and Hamiltonian NODEs alike misinterpret the bifurcation*, incorrectly identifying it as a symmetry-breaking transition, resulting in a spurious double-well structure. Notably, the double-well structure in the free-form model rapidly collapses with respect to $\xi$ (highlighted in purple), reverting to a normal saddle point (Figure 8 (c)), whereas the Hamiltonian model retains the misidentified structure more persistently (Figure 8 (d)).

This phenomenon can be explained using Proposition 4.1: both models successfully learn the local center-to-saddle bifurcation of the linear system. They also preserve an outer orbit structure that encapsulates the local training domain at $\xi \sim \xi_{\text{crit}}$. As a result, they hallucinate extra fixed points, mimicking broken symmetry. The Hamiltonian model, however, preserves the outer orbit more robustly due to its symplectic structure (Strogatz, 2018), leading to the persistent double-well configuration. In Appendix G, we also provide an experiment conducted under the standard pre-bifurcation training, which yielded the similar conclusion.

### 4.4. Diagnosing Hallucinated Bifurcations

Previous findings highlight the importance of determining whether a model hallucinates bifurcations. We propose a simple yet effective criterion: assessing the variance in bifurcation diagrams generated from multiple independent training runs, similar to the deep ensemble method (Lakshminarayanan et al., 2017). Figure 9 illustrates the estimated variance from 3-fold bootstrapped ensembles in two

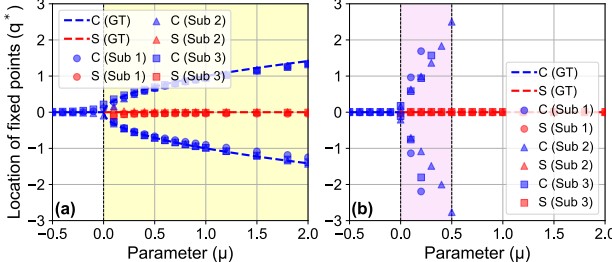

*Figure 9.* Ensembled bifurcation diagrams trained on (a) the double-well (Section 3.1) and (b) the linear system (Section 4.3). Each ensemble contains three sub-NODEs trained independently on bootstrapped datasets with different initializations.

scenarios: one involving a correctly identified bifurcation (from Section 3.1) and the other involving a hallucinated one (from earlier results). In Figure 9 (a), the symmetry-breaking is correctly captured, and the estimated variance remains minimal, indicating stable detection of the bifurcation. In contrast, the ground true bifurcation in Figure 9 (b) is a center-to-saddle at $q^* = 0$. However, the model incorrectly generates two centers of orbits within $0 < \mu < 0.5$, representing hallucinated behavior. This results in low variance ($5.343 \times 10^{-3}$ at $\mu = 0.2$) near the actual transition but significantly higher variance ($0.4605$ at $\mu = 0.2$) in the spurious region, reflecting its structural instability.

## 5. Applications to Broader Systems

### 5.1. Identifying the Cusp Catastrophe

Catastrophe theory (Thom, 1977; Zeeman, 1979) describes how small, continuous changes in parameters can lead to sudden, discontinuous shifts in system behavior. Among its five fundamental types (Saunders, 1980), the cusp catastrophe is one of the most well-known and extensively studied. The cusp catastrophe is a codimension-2 bifurcation system given by $\dot{\mathbf{x}} = (\dot{q}, \dot{p}) = (p, \nu + \mu q - q^3 - \kappa p)$, with parameters $(\mu, \nu)$ and fixed damping $\kappa = 0.5$. It is a simple extension of (3), but the inclusion of the $\nu$ term explicitly breaks symmetry, making it a canonical model for studying hysteresis phenomena. We investigated whether context-informed NODEs can identify the cusp bifurcation and hysteresis loop, despite being trained only on the pre-

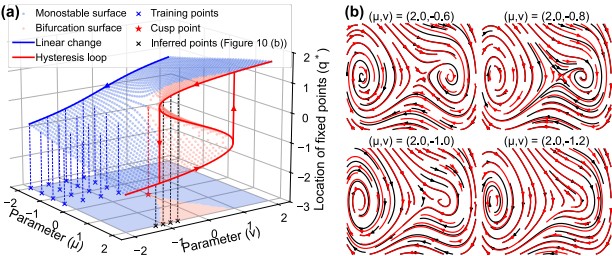

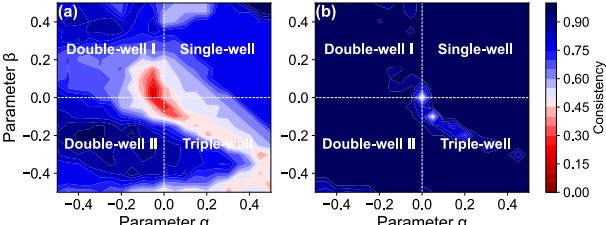

*Figure 10.* (a) Bifurcation surface generated by the context-informed model trained on a cusp bifurcation system. (b) Comparison between the phase portraits of the ground truth (black lines) and the learned model (red lines) near the catastrophic transition.

bifurcation regime. Specifically, training was conducted using parameters $(\mu_e^{\text{tr}}, \nu_e^{\text{tr}}) \in \{-2.0, -1.5, -1.0, -0.5\}^2$, which lie entirely within the monostable region. For each training parameters, four training trajectories were generated with $(q(0), p(0)) \sim \mathcal{U}([-2.0, 2.0]^2)$, $T = 2.0$ and $\Delta t = 0.1$. After training, the bifurcation surface was generated over a mesh grid $(\mu_e^{\text{test}}, \nu_e^{\text{test}}) \in [-2.0, 2.0]^2$, using the neural-adjoint method. Figures 10 (a) and (b) respectively show the identified bifurcation surface and the phase portraits of the learned vector fields near the catastrophic transition (see Appendix H for other cases). These results demonstrate that the model successfully captures the complicated bifurcation structure without explicitly learning it.

### 5.2. Identifying the Landau–Khalatnikov Theory

The LK theory (Landau & Khalatnikov, 1954) is a widely used mathematical framework for describing phase transition dynamics in various physical systems, particularly ferroelectric materials (Khan et al., 2015; Lo, 2003; Shi et al., 2016). The dynamics are governed by the following second-order ODE $\ddot{\psi} + \kappa\dot{\psi} = -\partial_\psi \mathcal{F}(\psi)$, where $\psi$ is an order parameter characterizes the phase of the system (e.g., polarization of ferroelectric materials), $\kappa$ is a fixed damping coefficient, and $\mathcal{F}(\psi)$ is the free energy expressed as a polynomial expansion $\mathcal{F}(\psi) = \frac{\alpha}{2}\psi^2 + \frac{\beta}{4}\psi^4 + \frac{\gamma}{6}\psi^6 + \cdots$. The coefficients $\alpha, \beta, \gamma, \ldots$ are phenomenological parameters typically expressed as functions of some physical quantities like temperature, e.g., $\alpha = \alpha(T - T_{\text{crit}})$. The phase transition of $\psi$ occurs as these parameters change with $T$. The LK model can be formulated in a Hamiltonian-like form as

$$(\dot{\psi}, \dot{\psi}') = \left(\psi', -\kappa\psi' - \alpha\psi - \beta\psi^3 - \gamma\psi^5 + \cdots\right), \quad (4)$$

except for the damping term $-\kappa\psi'$: if $\kappa \to 0$, then the system approaches precise Hamiltonian equations of motion.

The exact form of $\mathcal{F}$, such as how many orders should be included, is often unknown. However, one essential requirement is that $\mathcal{F}(\psi)$ must not diverge to negative infinity to preserve physical consistency. This constraint ensures that the trajectories of $(\psi, \psi')$ in (4) remain bounded, forming a closed outer orbit that encapsulates the system, especially

*Figure 11.* Consistency scores ($\sigma = 0.5$) for learning (4) with (a) vanilla and (b) regularized models, averaged over five runs.

for $\kappa \to 0$. In this case, the system must satisfy the conditions stated in Corollary 4.1. For nonzero $\kappa$, there is no strict guarantee that the total Poincaré index will always be constrained to $+1$. However, the introduction of a linear damping does not completely disrupt the overall phase topology. Consequently, the distribution of Poincaré indices is expected to be preserved. Inspired by this, we propose the following Poincaré–Hopf regularization:

$$\mathcal{R}_{\text{PH}}(\theta_c, W, \xi_e) = \|\text{Ind}(f(\cdot; \theta_c + W\xi_e), \Gamma_{\text{PH}}) - \chi_{\text{PH}}\|_2^2, \quad (5)$$

where $\Gamma_{\text{PH}}$ is a proper test contour and $\chi_{\text{PH}}$ is the desired index (see Appendix I for a detailed description of (5)).

We trained the conventional model and its regularized counterpart with (5) on the system (4). For training data, we simulated (4) over $T = 2.0$ with $\Delta t = 0.1$, using four initial conditions sampled from $(\psi(0), \psi'(0)) \sim \mathcal{U}([-2.0, 2.0]^2)$, for each combination of $(\alpha_e^{\text{tr}}, \beta_e^{\text{tr}}) \in \{-0.4, -0.2, 0.2, 0.4\}^2$. To ensure the physical relevance of (4), $\gamma = 0.05$ and $\kappa = 0.5$ were fixed. After training, we evaluated each model's ability to capture the long-term behavior and phase topology by assessing the consistency between its limit sets and those of the ground truth dynamics (Göring et al., 2024). Specifically, 32 trajectories were sampled and simulated during $T = 100.0$, for each parameter vector on a mesh grid over $(\alpha_e^{\text{test}}, \beta_e^{\text{test}}) \in [-0.5, 0.5]^2$. For each $(\alpha_e^{\text{test}}, \beta_e^{\text{test}})$, the corresponding $\xi_e^{\text{test}}$ was constructed using linear regression fitted on $(\alpha_e^{\text{tr}}, \beta_e^{\text{tr}})$, and then used as input to the model to simulate long-term trajectories also. If the ground truth and model dynamics converge within a distance of $\sigma$, we consider them consistent. Figure 11 confirms (5) enhances the long-term predictability across the parameter plane. Detailed results are in Appendix J, with exhaustive ablation studies of the proposed regularization in Appendix K.

## 6. Conclusions and Limitations

We show that NODEs can identify symmetry-breaking bifurcations solely from symmetric data, interpreted through the Poincaré index. We also propose a novel regularization inspired by the Poincaré–Hopf theorem. While the current method is limited to 2-dimensional flows with nearly closed orbits, Theorem 4.1 reveals that the concept can extend to a broad class of flows, paving the way for further research.

## Acknowledgements

We thank Jabir Bin Jahangir, Jisu Ryu and Young-Gu Kim for their helpful comments on an earlier draft of this work. Changwook Jeong acknowledges support by the National Research Foundation of Korea (NRF), funded by the Ministry of Science and ICT (MSIT) (RS-2023-00257666, RS-2024-00458251, RS-2024-00408180); by the Ministry of Trade, Industry and Energy (MOTIE) (RS-2023-00231956, P0023703); by the Institute for Information & Communications Technology Planning & Evaluation (IITP), funded by MSIT (RS-2020-II201336, AIGS); by the research project fund of UNIST (1.250004.01; 1.220104.01); and by Samsung Electronics. Muhammad Ashraful Alam acknowledges support by the Jai N. Gupta Distinguished Chair Professorship fund.

## Impact Statement

This paper explores the potential of NODE models in addressing OOD challenges in learning dynamics, focusing on bifurcation identification and generalization through the lens of topology and dynamical systems theory. The content is primarily theoretical, and no ethical concerns have been identified.

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

# A. A Brief Overview of the Poincaré–Hopf Index Theorem

This section provides a brief introduction to the Poincaré–Hopf index theory. For a more comprehensive treatment, we refer readers to standard textbooks on dynamical systems theory (Strogatz, 2018) and differential topology (Guillemin & Pollack, 2010; Milnor & Weaver, 1997).

**Definition A.1.** *(Poincaré index) Let $\mathcal{M}$ be an oriented smooth d-manifold and let $f : \mathcal{M} \to T\mathcal{M}$ be a smooth vector field. Suppose $\mathbf{x}^* \in \mathcal{M}$ is an isolated zero of f, i.e., $f(\mathbf{x}^*) = 0$ and there is a neighborhood U of $\mathbf{x}^*$ with $f(\mathbf{x}) \neq 0$ for all $\mathbf{x} \in U \setminus \{\mathbf{x}^*\}$. Choose an oriented coordinate chart around $\mathbf{x}^*$ and a closed d-ball $D \subset \mathcal{M}$ centered at $\mathbf{x}^*$ such that $\mathbf{x}^*$ is the only zero of f in D and $f(\mathbf{x}) \neq 0$ for all $\mathbf{x} \in \partial D$ on the boundary $\partial D$. The Poincaré index of f at $\mathbf{x}^*$, $\mathrm{Ind}(f, \mathbf{x}^*)$, is defined as the topological degree of the map:*

$$\Phi(\mathbf{x}) = \frac{f(\mathbf{x})}{\|f(\mathbf{x})\|} : \partial D \to \mathbb{S}^{d-1},$$

*that is,*

$$\mathrm{Ind}(f, \mathbf{x}^*) := \deg(\Phi) \in \mathbb{Z}.$$

*This integer is independent of the choice of coordinate chart and the ball D, and thus defines a topological invariant associated to the isolated zero $\mathbf{x}^*$. A visualization of the Poincaré index in $\mathbb{R}^2$ is shown in Figure 12.*

**Remark A.1.** *Intuitively, the map $\Phi(\mathbf{x}) = f(\mathbf{x})/\|f(\mathbf{x})\|$ normalizes the vector field, capturing only the direction of each vector and discarding its magnitude. Thus, $\Phi(\mathbf{x})$ maps each point $\mathbf{x}$ on $\partial D \cong \mathbb{S}^{d-1}$ to a point on the unit sphere $\mathbb{S}^{d-1}$, representing the direction of $f(\mathbf{x})$. The topological degree $\deg(\Phi)$ is an integer obtained by counting, with orientation signs, the preimages in $\partial D$ of any regular value in $\mathbb{S}^{d-1}$: $\deg(\Phi) = \sum_{\mathbf{x} \in \Phi^{-1}(y)} \mathrm{sign}(\det \mathcal{D}\Phi(\mathbf{x}))$ for a regular value $y \in \mathbb{S}^{d-1}$. This integer can be interpreted as an oriented wrapping number of $\partial D$ around the sphere $\mathbb{S}^{d-1}$ under the map $\Phi$. In this sense, Definition A.1 generalizes the classical winding number of a planar vector field in $\mathbb{R}^2$ to higher-dimensional cases.*

**Remark A.2.** *For an isolated, non-degenerate zero $\mathbf{x}^*$ of a vector field f, the Poincaré index can be computed as*

$$\mathrm{Ind}(f, \mathbf{x}^*) = \mathrm{sign}(\det \mathcal{D}f(\mathbf{x}^*)),$$

*where $\mathcal{D}f(\mathbf{x}^*)$ denotes the Jacobian matrix of f evaluated at $\mathbf{x}^*$. This means that the index of an isolated and non-degenerate zero must be either $+1$ or $-1$. Intuitively, the sign of the Jacobian determinant indicates whether the vector field locally preserves orientation ($\mathrm{Ind}(f, \mathbf{x}^*) = +1$) or reverses it ($\mathrm{Ind}(f, \mathbf{x}^*) = -1$) near the zero.*

**Remark A.3.** *In the language of differential topology, the topological degree of the map $\Phi : \partial D \to \mathbb{S}^{d-1}$ can be computed using the pullback of a volume form. Specifically,*

$$\deg(\Phi) \cdot \int_{\mathbb{S}^{d-1}} \omega = \int_{\partial D} \Phi^* \omega,$$

*where $\omega$ is the volume form on the unit sphere $\mathbb{S}^{d-1}$ and $\Phi^*\omega$ denotes its pullback via the map $\Phi$. Since the integral of the volume form over the unit $(d-1)$-sphere is simply given by $\int_{\mathbb{S}^{d-1}} \omega = \mathrm{Vol}(\mathbb{S}^{d-1}) = 2\pi^{d/2}/\Gamma(d/2)$, the Poincaré index can be computed as*

$$\mathrm{Ind}(f, \partial D) = \deg(\Phi) = \frac{1}{\mathrm{Vol}(\mathbb{S}^{d-1})} \int_{\partial D} \Phi^* \omega = \frac{\Gamma(d/2)}{2\pi^{d/2}} \int_{\partial D} \Phi^* \omega. \tag{6}$$

**Remark A.4.** *When evaluating (6), the expression is well-defined and yields a correct result as long as the test contour $\partial D$ does not pass through a zero of the vector field; that is, $f(\mathbf{x}) \neq 0$ for all $\mathbf{x} \in \partial D$. If multiple isolated zeros lie inside the region enclosed by $\partial D$, then (6) computes the total Poincaré index of all those zeros, defined as the sum of the indices of all enclosed zeros:*

$$\mathrm{Ind}(f, \partial D) = \sum_{\mathbf{x}^* \in \mathrm{int}(\partial D)} \mathrm{Ind}(f, \mathbf{x}^*).$$

*For $\mathcal{M} = \mathbb{R}^2$, this can be understood intuitively as follows: Suppose the contour $\partial D$ encloses several isolated zeros $\{\mathbf{x}_1^*, \mathbf{x}_2^*, \ldots, \mathbf{x}_k^*\}$. The contour $\partial D$ can be continuously deformed into a new closed curve $\partial D'$ consisting of k small loops $\{\partial D_1, \partial D_2, \ldots, \partial D_k\}$, each encircling exactly one of the zeros $\{\mathbf{x}_1^*, \mathbf{x}_2^*, \ldots, \mathbf{x}_k^*\}$, and connected by two-way narrow bridges. Since the contributions from these connecting bridges cancel out pairwise, the total index remains unchanged. This intuitive picture extends naturally to higher-dimensional cases, where the topological invariance of the index ensures that the total index depends only on the enclosed zeros, not on the specific shape of the test surface.*

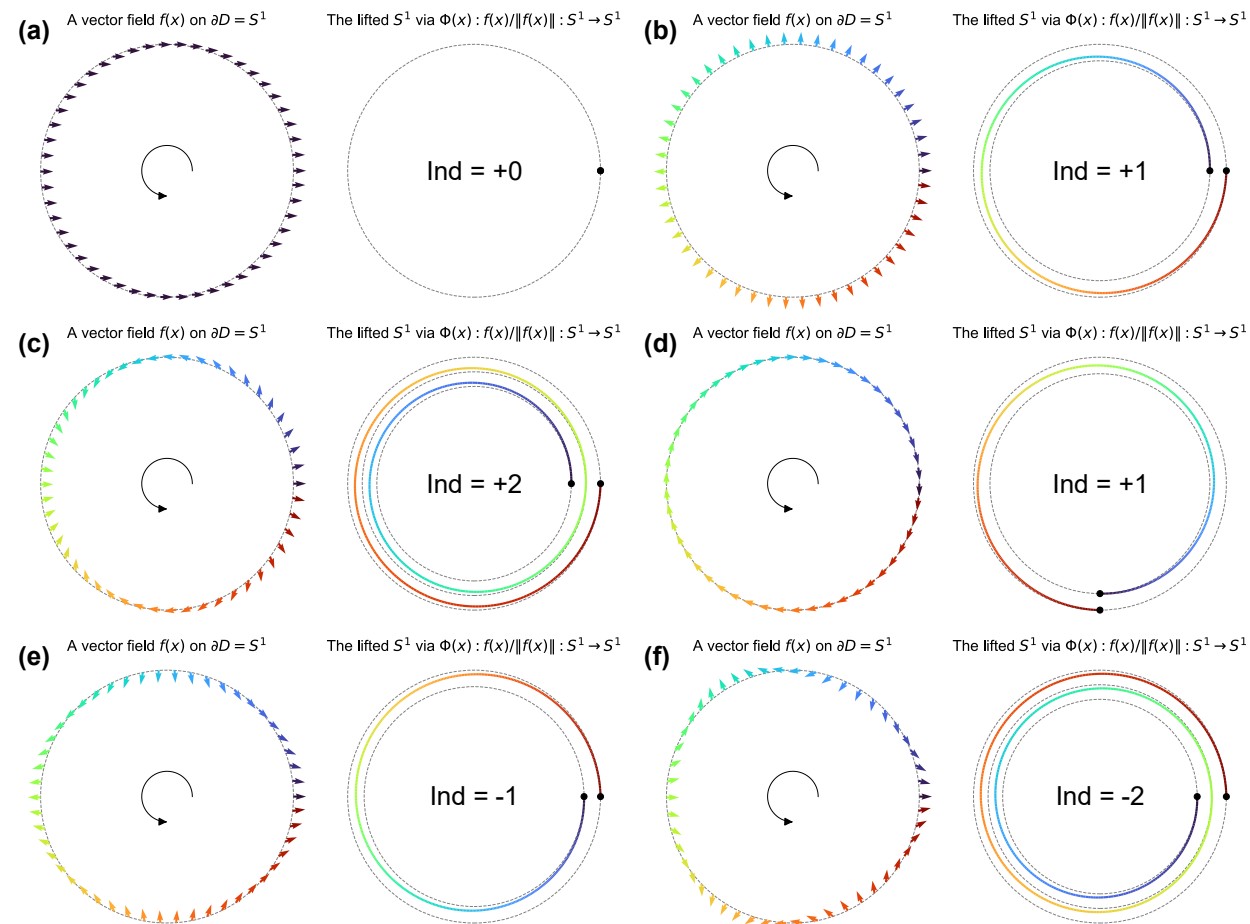

*Figure 12.* Visualization of six vector field configurations $f(\mathbf{x})$ on the circular test contour $\partial D = \mathbb{S}^1$ (left column), and their corresponding images on $\mathbb{S}^1$ via the map $\Phi(\mathbf{x}) = f(\mathbf{x})/\|f(\mathbf{x})\|$ (right column). The test contour $\partial D$ is traversed in the counterclockwise direction throughout. A color gradient from blue to red indicates the progression along the path. (a) A uniform vector field along $\partial D$ maps the entire circle to a single point on $\mathbb{S}^1$; the resulting Poincaré index is 0. (b) A radial vector field makes the contour wrap once around $\mathbb{S}^1$ in the counterclockwise direction, giving an index of $+1$. (c) Traversing $\partial D$ once counterclockwise, the vector field winds twice about the origin in the counterclockwise direction, so the image covers $\mathbb{S}^1$ twice and the index is $+2$. (d) A tangential vector field rotates once counterclockwise along the contour, again producing an index of $+1$. (e) A saddle-like configuration winds once clockwise relative to the counter-clockwise traversal, producing an index of $-1$. (f) A more complex field winds twice clockwise, resulting in an index of $-2$.

**Remark A.5.** *From the definition of the Poincaré index (6), its two-dimensional version can be computed as*

$$\mathrm{Ind}(f, \partial D) = \frac{1}{2\pi} \oint_{\partial D} \frac{-f_p \mathrm{d}f_q + f_q \mathrm{d}f_p}{f_q^2 + f_p^2}.$$

*While this expression can be derived directly from (6), it is more straightforward to obtain it using the standard definition of the winding number. For a real vector field $f(\mathbf{x} = (q, p)) = (f_q(\mathbf{x}), f_p(\mathbf{x}))$, let its complex representation be $F(\mathbf{x}) = f_q(\mathbf{x}) + i f_p(\mathbf{x})$. Because $\mathrm{d}\theta = \mathrm{Im}(\mathrm{d}F/F)$, the index can be written as*

$$\mathrm{Ind}(f, \partial D) = \frac{1}{2\pi} \oint_{\partial D} \mathrm{d}\theta = \frac{1}{2\pi} \oint_{\partial D} \mathrm{Im}\left(\frac{\mathrm{d}F}{F}\right) = \frac{1}{2\pi} \oint_{\partial D} \mathrm{Im}\left[\frac{(\mathrm{d}f_q + i\mathrm{d}f_p)(f_q - if_p)}{f_q^2 + f_p^2}\right] = \frac{1}{2\pi} \oint_{\partial D} \frac{-f_p \mathrm{d}f_q + f_q \mathrm{d}f_p}{f_q^2 + f_p^2}.$$

**Theorem A.1.** *(**Poincaré–Hopf Theorem**) Let $\mathcal{M}$ be a compact, oriented, smooth manifold without boundary, and let $f : \mathcal{M} \to T\mathcal{M}$ be a smooth vector field on $\mathcal{M}$ with finitely many isolated zeros $\{\mathbf{x}_1^*, \mathbf{x}_2^*, \ldots, \mathbf{x}_k^*\}$. Then, the sum of the Poincaré indices of $f$ at these zeros is equal to the Euler characteristic $\chi(\mathcal{M})$ of $\mathcal{M}$:*

$$\sum_{i=1}^{k} \mathrm{Ind}(f, \mathbf{x}_i^*) = \chi(\mathcal{M}).$$

**Corollary A.1.** (*Poincaré–Hopf for Closed Orbits*) *Let $f : \mathbb{R}^2 \to \mathbb{R}^2$ be a smooth vector field and let $\Gamma$ be a simple (non-self-intersecting) closed orbit of the dynamical system $\dot{\mathbf{x}} = f(\mathbf{x})$. Suppose that all fixed points (all zeros of $f$) inside $\Gamma$ are isolated. Then, the sum of the Poincaré indices of all fixed points inside $\Gamma$ is equal to $+1$:*

$$\sum_{\mathbf{x}^* \in \text{int}(\Gamma)} \text{Ind}(f, \mathbf{x}^*) = +1.$$

*Proof.* Because $\mathcal{M} = \mathbb{R}^2$ and $\Gamma$ is an actual trajectory of the dynamical system $\dot{\mathbf{x}} = f(\mathbf{x})$, $f(\mathbf{x})$ is tangential at every $\mathbf{x} \in \Gamma$. It ensures $\text{Ind}(f, \Gamma)$ is equal to $+1$. Then, from Remark A.4, $\text{Ind}(f, \Gamma) = \sum_{i=1}^{k} \text{Ind}(f, \mathbf{x}_i^*) = +1$, where $\mathbf{x}_1^*, \mathbf{x}_2^*, \ldots, \mathbf{x}_k^*$ are fixed points lying in the interior of $\Gamma$. □

# B. Mathematical Details and Proofs

**Assumption B.1.** *Consider a separable Hamiltonian $\mathcal{H} : \mathbb{R}^2 \times \mathbb{R} \to \mathbb{R}$:*

$$\mathcal{H}(q, p; \mu) = K(p) + V_\mu(q) = \frac{p^2}{2} - \frac{\mu q^2}{2} - \int \mathcal{P}(q) \mathrm{d}q + C,$$

*where $\mathcal{P} : \mathbb{R} \to \mathbb{R}$ is a smooth function satisfying $\mathcal{P}(0) = \partial_q \mathcal{P}(0) = 0$. The ground truth vector field $f : \mathbb{R}^2 \times \mathbb{R} \to \mathbb{R}^2$ is assumed to be a Hamiltonian vector field of $\mathcal{H}$ and expressed as*

$$(\dot{q}, \dot{p}) = f(q, p; \mu) = \left( \frac{\partial \mathcal{H}(q, p; \mu)}{\partial p}, -\frac{\partial \mathcal{H}(q, p; \mu)}{\partial q} \right) = (p, \mu q + \mathcal{P}(q)),$$

**Remark B.1.** *The Hamiltonian vector field $f$ defined in Assumption B.1 has an isolated fixed point at $(0,0)$ because $f(0,0; \mu) = (0, \mathcal{P}(0)) = (0,0)$. Note that the Jacobian of $f$ at $(0,0)$ is*

$$\mathcal{D}f(q, p; \mu) = \begin{pmatrix} \partial_q \dot{q} & \partial_p \dot{q} \\ \partial_q \dot{p} & \partial_p \dot{p} \end{pmatrix} = \begin{pmatrix} 0 & 1 \\ \mu + \partial_q \mathcal{P}(q) & 0 \end{pmatrix} \implies \mathcal{D}f(0,0; \mu) = \begin{pmatrix} 0 & 1 \\ \mu & 0 \end{pmatrix}.$$

*The characteristic polynomial is*

$$\chi_f(\lambda, \mu) = \det(\mathcal{D}f(0,0; \mu) - \lambda I) = \lambda^2 - \mu = 0 \implies \lambda = \pm\sqrt{\mu}.$$

*Therefore, the fixed point $(0,0)$ is a center for $\mu < 0$ ($\lambda = \pm i\sqrt{|\mu|}$) and a saddle for $\mu > 0$ ($\lambda = \pm\sqrt{|\mu|}$), which means the local bifurcation is the center-to-saddle type.*

**Remark B.2.** *Consider $\mathcal{P}(q) = 0$. It satisfies Assumption B.1. In this case, $(0,0)$ is the only fixed point. The system exhibits a center-to-saddle bifurcation at $(0,0)$ simply.*

**Remark B.3.** *Consider $\mathcal{P}(q) = -q^3$. It satisfies Assumption B.1. The fixed points of $f$ are determined as follows:*

$$f(q, p; \mu) = (p, q(\mu - q^2)) = (0,0) \implies (q, p) = (0,0), (\pm\sqrt{\mu}, 0).$$

*The additional fixed points $(q, p) = (\pm\sqrt{\mu}, 0)$ exist only for $\mu > 0$. Thus, the system undergoes a $1 \mapsto 3$ bifurcation at $\mu = 0$, generating two additional fixed points. For these fixed points, the Jacobian is given by*

$$\mathcal{D}f(q, p; \mu) = \begin{pmatrix} 0 & 1 \\ \mu - 3q^2 & 0 \end{pmatrix} \implies \mathcal{D}f(\pm\sqrt{\mu}, 0; \mu) = \begin{pmatrix} 0 & 1 \\ -2\mu & 0 \end{pmatrix},$$

*The characteristic polynomial for these fixed points is*

$$\chi_f(\lambda, \mu) = \lambda^2 + 2\mu = 0 \implies \lambda = \pm\sqrt{-2|\mu|},$$

*where $\mu > 0$. Thus, the emerging fixed points are center-like, and the system exhibits a symmetry-breaking bifurcation.*

**Lemma B.1.** *Any fixed point of a Hamiltonian system in a 2-dimensional phase space must either be a center of closed orbits or a saddle point.*

*Proof.* It is a direct application of the volume-preserving property of Hamiltonian dynamics (Arnold, 2013). The divergence of a Hamiltonian system holds

$$\nabla \cdot f(q, p) = \frac{\partial \dot{q}}{\partial q} + \frac{\partial \dot{p}}{\partial p} = \frac{\partial}{\partial q}\frac{\partial \mathcal{H}}{\partial p} - \frac{\partial}{\partial p}\frac{\partial \mathcal{H}}{\partial q} = 0,$$

for any $(q, p)$ and $\mu$, regardless of the specific form of $\mathcal{H}$. This implies that the trace of the Jacobian must be zero:

$$\operatorname{tr}(\mathcal{D}f(q, p)) = \nabla \cdot f(q, p) = 0.$$

This ensures that the characteristics equation at $(q, p)$ is given by

$$\chi_f(\lambda, \mu) = \lambda^2 + \det\left(\mathcal{D}f(q, p)\right).$$

Consequently, the fixed points $(q, p)$ of a Hamiltonian system must fall into one of the following two categories: a center when $\det\left(\mathcal{D}f(q, p)\right) > 0$ or a saddle when $\det\left(\mathcal{D}f(q, p)\right) < 0$. Moreover, in the case of separable Hamiltonians, $\det\left(\mathcal{D}f(q, p)\right) = \partial_q^2 \mathcal{H}(q, p) = \partial_q^2 V(q)$. $\square$

**Lemma B.2.** *Let $g : \mathbb{R}^2 \times \mathbb{R} \to \mathbb{R}^2$ is a smooth vector field, Assume there exists a smooth bijective map $\phi : \mathbb{R} \to \mathbb{R}$ such that $f$ and $g$ are $\delta$-close in the $\mathcal{C}^1$ sense:*

$$\sup_{\mathbf{x} \in U}\|g(\mathbf{x}; \phi(\mu)) - f(\mathbf{x}; \mu)\|_2 + \sup_{\mathbf{x} \in U}\|\mathcal{D}g(\mathbf{x}; \phi(\mu)) - \mathcal{D}f(\mathbf{x}; \mu)\|_2 \leq \delta,$$

*in some open interval $\mu \in (-\epsilon, \epsilon)$, where $f$ is as defined in Assumption B.1 and $U \in \mathbb{R}^2$ is some neighborhood of $(0, 0)$. Then, there exists a fixed point $\mathbf{x}^* \simeq (0, 0)$ which is a near-center when $\mu < \mu_{\text{crit}} \simeq 0$ and a saddle when $\mu > \mu_{\text{crit}} \simeq 0$.*

*Proof.* It follows directly from (Crawford, 1991). From the statement, we can rewrite $g$ as

$$g(\mathbf{x}; \phi(\mu)) = f(\mathbf{x}; \mu) + \Delta(\mathbf{x}; \mu), \quad \|\Delta(\mathbf{x}; \mu)\| \leq \mathcal{O}(\delta).$$

Similarly, the Jacobians are given by

$$\mathcal{D}g(\mathbf{x}; \phi(\mu)) = \mathcal{D}f(\mathbf{x}; \mu) + \mathcal{D}\Delta(\mathbf{x}; \mu), \quad \|\mathcal{D}\Delta(\mathbf{x}; \mu)\| \leq \mathcal{O}(\delta).$$

First, we want to show that for each small $\mu \neq 0$, there exists a fixed point of $g(\cdot, \phi(\mu))$ closed to $\mathbf{x} = (0, 0)$. Note that

$$g(\mathbf{0}; \phi(\mu)) = f(\mathbf{0}; \mu) + \Delta(\mathbf{0}; \mu) = \Delta(\mathbf{0}; \mu).$$

By denoting $G(\mathbf{x}, \mu) = g(\mathbf{x}; \phi(\mu))$, we have $\|G(\mathbf{0}, \mu)\| \leq C\delta$. Consider the Jacobians at $\mathbf{x} = (0, 0)$:

$$\mathcal{D}g(\mathbf{0}; \phi(\mu)) = \mathcal{D}f(\mathbf{0}; \mu) + \mathcal{D}\Delta(\mathbf{0}; \mu).$$

Because $\det(\mathcal{D}f(\mathbf{0}; \mu)) = -\mu$, $\mathcal{D}f(\mathbf{0}; \mu)$ is invertible for all $\mu \neq 0$. Then, for a sufficiently small $\delta$,

$$\det |\mathcal{D}g(\mathbf{0}; \phi(\mu))| = -\mu + \mathcal{O}(\delta) \neq 0,$$

thus $\mathcal{D}g(\mathbf{x}; \mu)$ remains invertible at $\mathbf{x} = 0$ and $\mu \neq 0$. Therefore, from the given conditions

$$\|G(\mathbf{0}, \mu)\| \leq C\delta, \quad \det(\mathcal{D}G(\mathbf{0}; \mu)) \neq 0,$$

the continuity and implicit function theorem imply that for $\mu \neq 0$ there exists an unique solution $G(\mathbf{x}(\delta, \mu), \mu) = g(\mathbf{x}(\delta, \mu), \phi(\mu)) = \mathbf{0}$ satisfying $\mathbf{x}(\delta, \mu) \to (0, 0)$ as $\delta \to 0$. It directly gives $g$ has an isolated fixed point $\mathbf{x}(\delta, \mu)$ close to $(0, 0)$ whenever $\mu \neq 0$.

We now show that $(\mathbf{x}(\delta, \mu), \mu)$ of $g$ has a unique smooth curve $(\lambda, \mu)$ of eigenvalues near $\lambda_0$ for $\mu$ in a neighborhood of $\mu_0 \neq 0$. Consider the Jacobian at $\mathbf{x}(\delta, \mu)$:

$$\mathcal{D}g(\mathbf{x}(\delta, \mu); \phi(\mu)) = \mathcal{J}_g(\delta, \mu) \simeq \mathcal{D}f(\mathbf{0}; \mu) + \mathcal{D}\Delta(\mathbf{x}(\delta, \mu); \mu) = \mathcal{J}_f(\mu) + \mathcal{J}_\Delta(\delta, \mu).$$

The characteristic polynomial of $\mathcal{J}_g(\delta, \mu)$ is given by

$$\chi_g(\lambda, \mu, \delta) = \lambda^2 - \mu - \operatorname{tr}(\mathcal{J}_\Delta(\delta, \mu))\lambda + \mathcal{O}(\delta^2) = \chi_f(\lambda, \mu) - \operatorname{tr}(\mathcal{J}_\Delta(\delta, \mu))\lambda,$$

where $\text{tr}(\mathcal{J}_\Delta) \leq \mathcal{O}(\delta)$. For $(\lambda, \mu) = (\lambda_0, \mu_0)$ satisfying $\chi_f(\lambda_0, \mu_0) = 0$ and $(\lambda_0, \mu_0) \neq (0, 0)$, we have

$$\chi_g(\lambda_0, \mu_0, \delta) = -\text{tr}(\mathcal{J}_\Delta(\delta, \mu_0))\lambda_0,$$

thus $|\chi_g(\lambda_0, \mu_0, \delta)| \leq C\delta$. Then, consider the partial derivative of $\chi_g$ with respect to $\lambda$ at $(\lambda_0, \mu_0)$

$$\frac{\partial \chi_g}{\partial \lambda}(\lambda_0, \mu_0, \delta) = 2\lambda - \text{tr}(\mathcal{J}_\Delta(\delta, \mu)),$$

which remains nonzero for a sufficiently small $\delta$. Observe the given conditions

$$|\chi_g(\lambda_0, \mu_0, \delta)| \leq C\delta, \quad \left| \frac{\partial \chi_g}{\partial \lambda}(\lambda_0, \mu_0, \delta) \right| = 2\lambda - C\delta > 0.$$

Again, by the continuity and implicit function theorem, there exists a unique curve $(\lambda(\delta), \mu(\delta))$ such that $\chi_g(\lambda(\delta), \mu(\delta)) = 0$, $(\lambda(\delta), \mu(\delta)) \to (\lambda_0, \mu_0)$ as $\delta \to 0$ for $\mu \neq 0$ and a sufficiently small $\delta$.

Now consider $\mu = -|\mu| < 0$ case. From the characteristic equation $\chi_g$ discussed above,

$$\chi_g(\lambda, \mu, \delta) = \lambda^2 + |\mu| - \text{tr}(\mathcal{J}_\Delta(\delta, \mu))\lambda \implies \lambda = \frac{\text{tr}(\mathcal{J}_\Delta(\delta, \mu))}{2} \pm i\sqrt{|\mu|},$$

thus $\text{Re}(\lambda) \leq \mathcal{O}(\delta)$. Therefore, for $\mu < 0$, the fixed point $\mathbf{x}^* \simeq \mathbf{0}$ of $g$ behaves like a center, although it may exhibit infinitesimal growth or damping depending on the sign of $\text{tr}(\mathcal{J}_\Delta)$. Next, for $\mu = |\mu| > 0$, we have

$$\chi_g(\lambda, \mu, \delta) = \lambda^2 - |\mu| - \text{tr}(\mathcal{J}_\Delta(\delta, \mu))\lambda \implies \lambda = \frac{\text{tr}(\mathcal{J}_\Delta(\delta, \mu))}{2} \pm \sqrt{|\mu|}.$$

Given that a small perturbation cannot alter the signs of $\lambda$, the fixed point $\mathbf{x}^* \simeq \mathbf{0}$ of $g$ is classified as a saddle point for $\mu > 0$, though the rate of convergence or divergence may vary slightly along each eigenvector direction. $\qquad\square$

**Proposition B.1.** *Let $f$ and $g$ satisfy the assumptions stated in Lemma B.2. In addition, suppose that the dynamical system $\dot{\mathbf{x}} = g(\mathbf{x}; \xi = \phi(\mu))$ admits at least one closed orbit $\Gamma$ that encloses an isolated fixed point $\mathbf{x}_0^* \simeq (0, 0)$ for a neighborhood of $\phi(\mu_{\text{crit}})$. Then, $g$ undergoes a (generalized) symmetry-breaking bifurcation, at least locally near $\xi_{\text{crit}}$.*

*Proof.* From Lemma B.2, the system $\dot{\mathbf{x}} = g(\mathbf{x}, \phi(\mu))$ has a fixed point $\mathbf{x}_0^*(\mu)$ near $(0, 0)$, and undergoes a center-to-saddle bifurcation. It means that, as $\mu$ passes through $\mu_{\text{crit}} \simeq 0$, the Poincaré index at $\mathbf{x}_0^*(\mu)$, $\text{Ind}(g, \mathbf{x}_0^*(\mu))$ changes from $\text{Ind}(g, \mathbf{x}_0^*(\mu < \mu^{\text{crit}})) = +1$ (a center) to $\text{Ind}(g, \mathbf{x}_0^*(\mu > \mu_{\text{crit}})) = -1$ (a saddle), resulting in a net index change of $-2$.

By assumption, there exists a closed orbit $\Gamma$ encircling $\mathbf{x}^*(\mu)$ for a neighborhood of $\phi(\mu_{\text{crit}})$. According to Corollary A.1, the sum of Poincaré indices of all fixed points enclosed by $\Gamma$ must be $+1$ near $\mu = \mu_{\text{crit}}$. Consequently, to preserve the total index of $+1$ within $\Gamma$, there must emerge additional fixed points, $\mathbf{x}_1^*, \ldots, \mathbf{x}_k^*$, within $\Gamma$, whose combined Poincaré indices sum to $+2$:

$$\sum_{i=1}^k \text{Ind}(f, \mathbf{x}_i^*) = +2.$$

As established in Lemma B.1, $f$ is a Hamiltonian system, thus its divergence $\nabla \cdot f(\mathbf{x}; \mu) = 0$ for all $\mathbf{x}$ and $\mu$. This implies that the trace of the Jacobian $\text{tr}(\mathcal{D}f(\mathbf{x}; \mu)) = 0$ for all $\mathbf{x}$ and $\mu$. This constraint ensures that the eigenvalues and accordingly the types of fixed points must fall into one of the following two categories: (1) purely imaginary (a center) or (2) real with opposite signs (a saddle).

Then, as established in Lemma B.2, because $g$ is $\delta$-close to $f$ in the $\mathcal{C}^1$ sense, it ensures $\|\nabla \cdot g(\mathbf{x}; \mu)\| = |\text{tr}(\mathcal{D}g(\mathbf{x}; \mu))| \leq \mathcal{O}(\delta)$ within $U$. For sufficiently small $\delta$, this ensures that no fixed points with large real parts in their eigenvalues can spontaneously arise. Consequently, any non-saddle fixed points of $g$ in $U$ must be near-centers, retaining a Poincaré index of $+1$ by continuity. Thus, under a generic scenario, the newly created fixed points $\mathbf{x}_1^*, \ldots, \mathbf{x}_k^*$ are two centers, contributing a total index of $+2$. $\qquad\square$

**Definition B.1.** *(Parameter OOD) Let $\mathcal{P} \subset \mathbb{R}^n$ be a parameter space and let $\mathcal{B} \subset \mathcal{P}$ be a bifurcation set. The complement $\mathcal{P} \setminus \mathcal{B}$ admits a decomposition into $L$ disjoint connected components: $\mathcal{P} \setminus \mathcal{B} = \bigcup_{i=1}^L \mathcal{P}_i$, where each $\mathcal{P}_i$ is a maximal connected subdomain. Intuitively, each $\mathcal{P}_i$ is one qualitatively uniform parameter regime. Then, a parameter OOD condition in learning dynamics arises when the support of training distribution is $\text{supp}(p_e^{\text{tr}}(\mu)) \subseteq \mathcal{P}_l$, but the support of the test distribution $\text{supp}(p_e^{\text{test}}(\mu)) \subseteq \mathcal{P}_{i \neq l}$. Equivalently, there exists no continuous path $\gamma : [0, 1] \to \mathcal{P} \setminus \mathcal{B}$ connecting $\mu_e^{\text{tr}}$ and $\mu_e^{\text{test}}$ without crossing $\mathcal{B}$.*

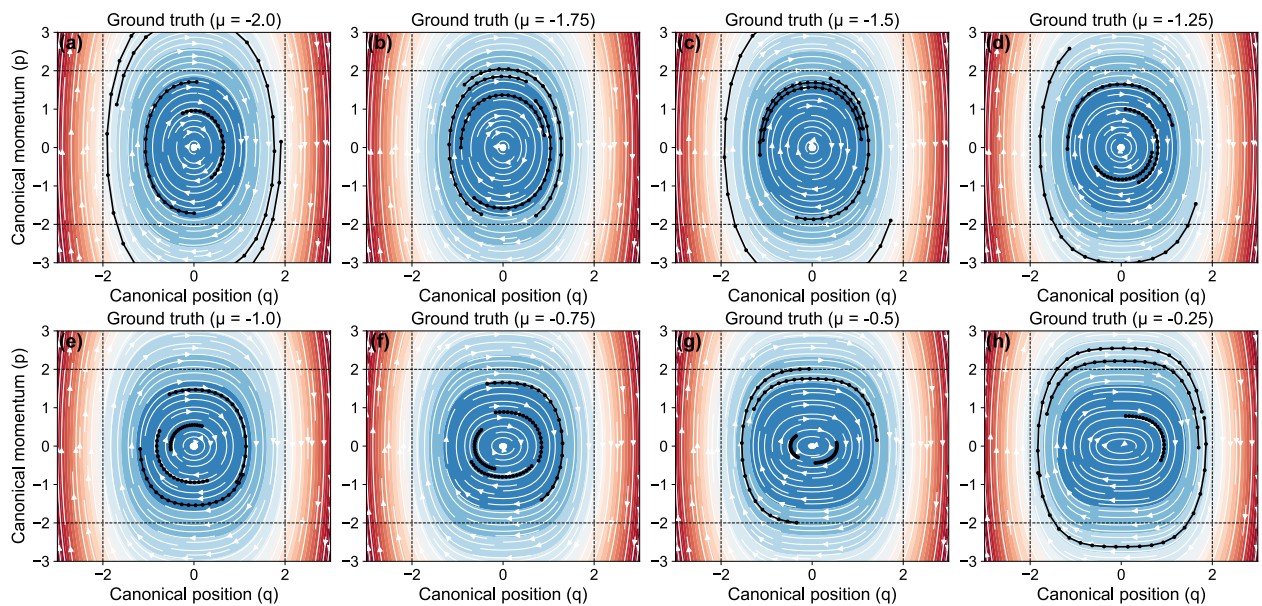

*Figure 13.* Examples of training trajectories from the experiment described in Section 3.1.

## C. Automatic Generation of Bifurcation Diagrams

To automatically generate the bifurcation diagram, the fixed points of the learned context-informed NODE must be determined for each variable $\xi$. This corresponds exactly to solving the inverse problem of $f(\mathbf{x}; \xi) = \mathbf{0}$ given by

$$\mathbf{x}^* = f^{-1}(\mathbf{0}; \xi),$$

for a given $\xi$. Because $f$ is fully differentiable, the neural-adjoint method (Ren et al., 2020) provides one of the simplest solutions to this inverse problem. The neural-adjoint method employs gradient descent to iteratively update the candidate $\mathbf{x}^i$ over $K$ iterations, aiming to satisfy $f(\mathbf{x}^K; \xi) = \mathbf{0}$:

$$\mathbf{x}^{i+1} = \mathbf{x}^i - \eta \nabla_{\mathbf{x}} \mathcal{L}(f(\mathbf{x}^i; \xi), \mathbf{0}) = \mathbf{x}^i - \eta \nabla_{\mathbf{x}} \|f(\mathbf{x}^i; \xi)\|_2^2,$$

where $\eta$ is a learning rate and $\mathcal{L}(f(\mathbf{x}), \mathbf{y})$ is a conventional loss function, such as MSE, designed for inverse problems of the form $f^{-1}(\mathbf{y}) = \mathbf{x}$. In this case, since we aim to find fixed points, $\mathbf{y} = \mathbf{0}$ and the loss function simplifies to the squared $L_2$ norm. This procedure is performed on a batch of size $N$ with initial values $\{\mathbf{x}_j^0\}_{j=1}^N, \mathbf{x}^0 \sim p(\mathbf{x}^0)$, where $p$ is a prior distribution, typically defined as a uniform distribution $U(\mathcal{N})$ over an initial domain $\mathcal{N}$. In our experiment, we set $K = 1,000, \eta = 10^{-2}, N = 100$, and $p(\mathbf{x}^0) = \mathcal{U}([-2.0, 2.0] \times [-2.0, 2.0])$.

After completing the iteration procedure ($i = K$), we discard points where the squared $L_2$ norm $\|f(\mathbf{x}^K; \xi)\|_2^2$ exceeds a threshold $\epsilon$. For the remaining candidates $\mathbf{x}^K$, we compute the Jacobians $\mathcal{D}f(\mathbf{x}^K; \xi)$. Using the eigenvalues of $\mathcal{D}f(\mathbf{x}^K; \xi)$, we classify the fixed point candidates based on linearization theory (Strogatz, 2018). Specifically, if the eigenvalues $\lambda_1$ and $\lambda_2$ are both real with opposite signs, the fixed point is classified as a saddle point. If $\lambda_1$ and $\lambda_2$ are purely imaginary and have opposite signs, the fixed point is classified as a center of orbits[4], and so on. For further details on the classification rule, refer to (Strogatz, 2018). After classifying all found fixed points, if any two points within a distance of $\delta$ have the same classification label, then they are merged by averaging. We used $\epsilon = 10^{-8}$ and $\delta = 10^{-2}$ for the experiments.

## D. Experiment Details: Section 3.1

**Data preparation.** We trained the context-informed NODEs (2) on the Hamiltonian system described in Remark B.3 under the pre-bifurcation regime exclusively. Specifically, we randomly sampled four initial conditions from the uniform distribution $(q(0), p(0)) \sim \mathcal{U}([-2.0, 2.0] \times [-2.0, 2.0])$ for each of the eight parameter values $\mu_e^{\text{tr}} \in \{-2.0, -1.75, -1.5, -1.25, -1.0, -0.75, -0.5, -0.25\}$. For each sampled initial condition, we simulated the dynamics with a time horizon $T = 2.0$ and a time step $\Delta t = 0.1$. This results in $|D_e| = 4$ trajectories per value of $\mu_e^{\text{tr}}$, yielding a total of $|D_e| \times 8 = 32$ training

---

[4]In practice, a fixed point with complex eigenvalues dominated by imaginary parts of opposite signs is regarded as center-like.

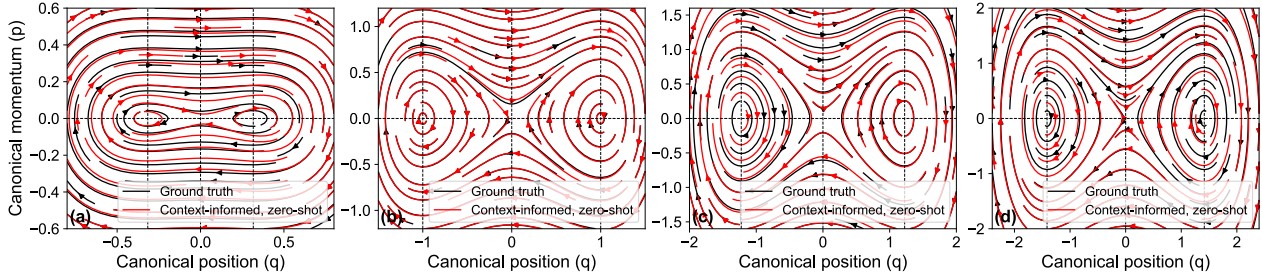

*Figure 14.* Predicted phase portraits of the (a) vanilla NODE and (b) context-informed NODE models for the right well adaptation scenario. Predicted phase portraits of the (c) vanilla NODE and (d) context-informed NODE models for the outer orbit adaptation scenario.

*Figure 15.* Phase portraits constructed in a zero-shot manner by linearly extrapolating the context vector $\xi_e$ based on pre-bifurcation training data, corresponding to different values of $\mu_e$: (a) $\mu_e = 0.1$, (b) $\mu_e = 1.0$, (c) $\mu_e = 1.5$, and (d) $\mu_e = 2.0$.

trajectories. It is important to note that all training data consists solely of single-orbit trajectories from the pre-bifurcation symmetric regime. The phase portraits of the training trajectories are visualized in Figure 13.

**Architecture.** We basically followed the settings outlined in the original CoDA paper (Kirchmeyer et al., 2022). We employed 4-layer multi-layer perceptrons (MLPs) with hidden layers of width 64 and `swish` activation functions (Ramachandran et al., 2017). We used the fourth-order Runge-Kutta (RK) method as an ODE solver.

**Training details.** We basically followed the settings outlined in the original CoDA paper (Kirchmeyer et al., 2022). We trained the context-informed NODEs using the Adam optimizer (Kingma & Ba, 2015) with default settings for 50,000 epochs, using a learning rate of $10^{-3}$ and a full-batch size. To improve training stability, we employed exponential scheduled sampling for teacher forcing (Lamb et al., 2016), starting with an initial probability of 0.99 and applying a decay rate of 0.99 every 30 epochs. $\lambda_\xi = 10^{-4}$ and $\lambda_\Omega = 10^{-6}$ were used for the sparsity regularizer $\mathcal{R}(W, \xi_e)$ in (2).

For the one-shot adaptation, We adapted the context-informed NODEs using the Adam optimizer with default settings for 1,000 epochs, using a learning rate of $10^{-3}$ and a full-batch size. We employed exponential scheduled sampling for teacher forcing, starting with an initial probability of 0.95 and applying a decay rate of 0.95 every 30 epochs. In comparison, the vanilla counterpart was trained for 3,000 epochs under the same settings.

**One-shot adaptation.** After training the model with pre-bifurcation data, we adapted it using a single trajectory for $\mu = 0.5$, representing post-bifurcation data. We considered the following four *broken symmetry* scenarios, namely adaptations using (i) a trajectory confined to the left well, (ii) a trajectory confined to the right well, and (iii) a trajectory outside the separatrix, traversing the outer orbit. These scenarios limit the model's exposure to the global structure of the phase space during adaptation, reflecting the spontaneous symmetry breaking observed in real-world situations. For comparison, we also trained vanilla neural ODE models (1) under each scenario. Figure 14 compares the phase portraits of vanilla NODEs and context-informed NODEs: (a–b) for the right well adaptation scenario and (c–d) for the outer orbit adaptation scenario (see Figure 2 for the left well scenario). The context-informed NODEs successfully reconstruct the entire phase topology across all scenarios.

**Zero-shot exploration.** We extrapolated the context values for post-bifurcation cases with $\mu_e = 0.1$, $\mu_e = 1.0$, $\mu_e = 1.5$, and $\mu_e = 2.0$ using a linear regression model fitted to the relationship between pre-bifurcation $\xi_e$ and $\mu_e$ (see Figure 3 (a)). Figure 15 illustrates the resultant phase portraits obtained in a zero-shot manner, showing that this method effectively reproduces the ground truth phase portraits with reasonable accuracy.

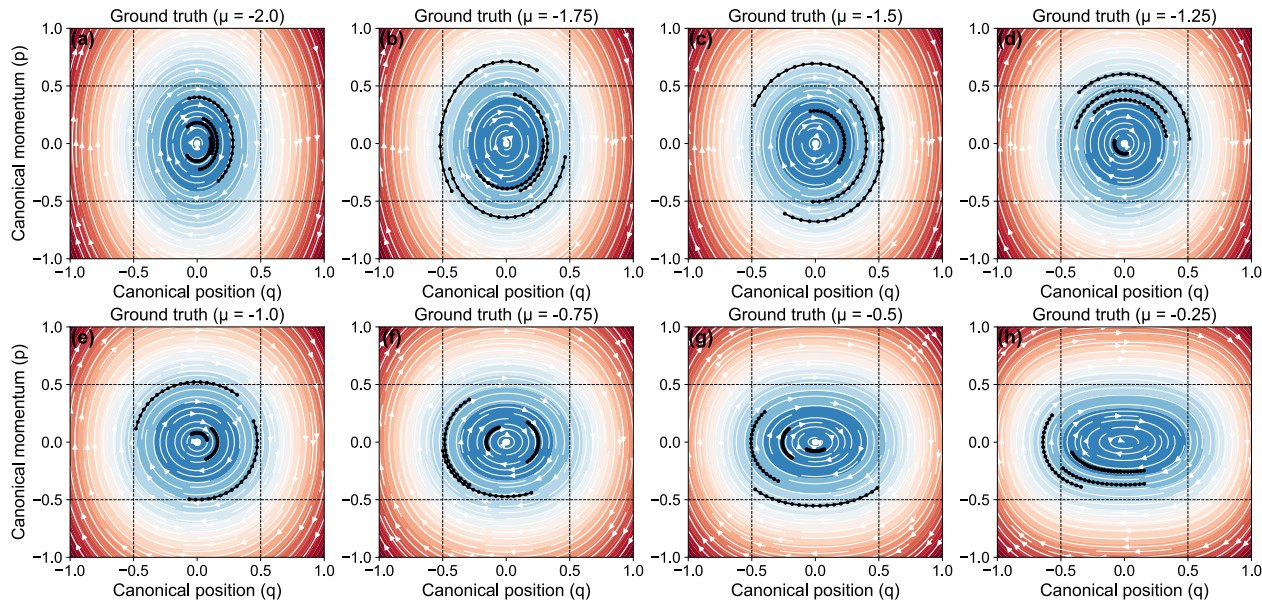

*Figure 16.* Examples of training trajectories from the experiment described in Section 3.2.

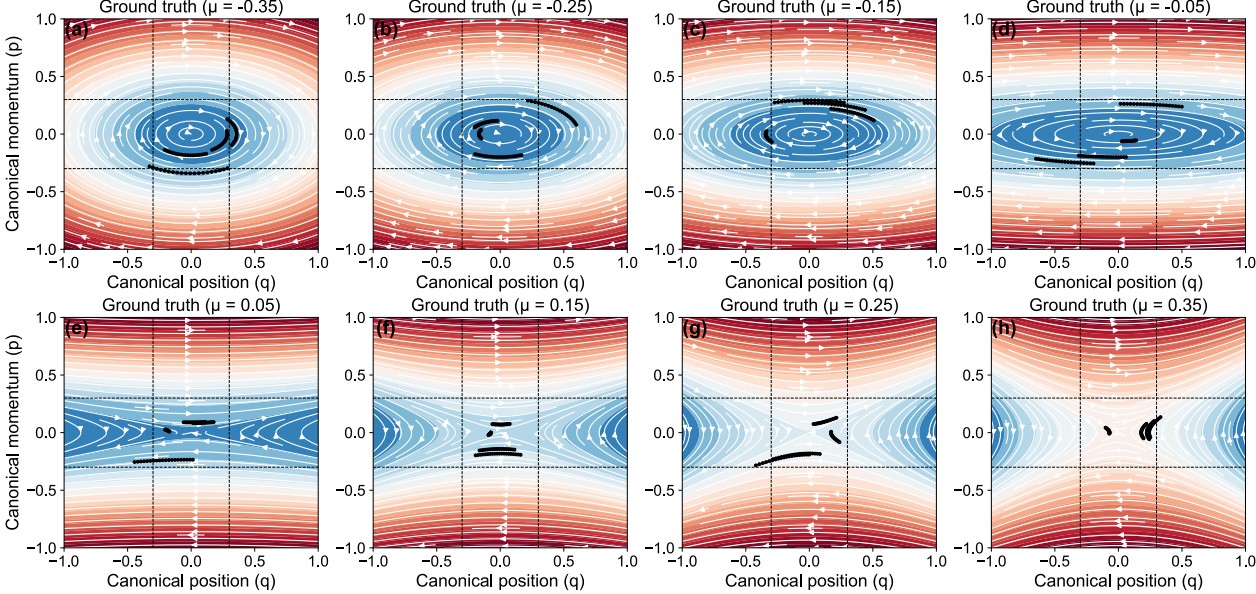

*Figure 17.* Examples of training trajectories from the experiment described in Section 4.3.

# E. Experiment Details: Section 3.2

**Data preparation.** We trained the context-informed NODEs (2) on the Hamiltonian system described in Remark B.3 under the pre-bifurcation regime with a restricted training domain. Specifically, we randomly sampled four initial conditions from the uniform distribution $(q(0), p(0)) \sim \mathcal{U}([-0.5, 0.5] \times [-0.5, 0.5])$ for each of the eight parameter values $\mu_e^{\mathrm{tr}} \in \{-2.0, -1.75, -1.5, -1.25, -1.0, -0.75, -0.5, -0.25\}$. For each sampled initial condition, we simulated the dynamics with a time horizon $T = 2.0$ and a time step $\Delta t = 0.1$. This results in $|D_e| = 4$ trajectories per value of $\mu_e^{\mathrm{tr}}$, yielding a total of $|D_e| \times 8 = 32$ training trajectories. It is important to note that all training data consists solely of single-orbit trajectories from the pre-bifurcation symmetric regime. The phase portraits of the training trajectories are visualized in Figure 16.

**Architecture.** We basically followed the settings outlined in the original CoDA paper (Kirchmeyer et al., 2022). We employed 4-layer MLPs with hidden layers of width 64 and `swish` activation functions. We used the fourth-order RK method as an ODE solver.

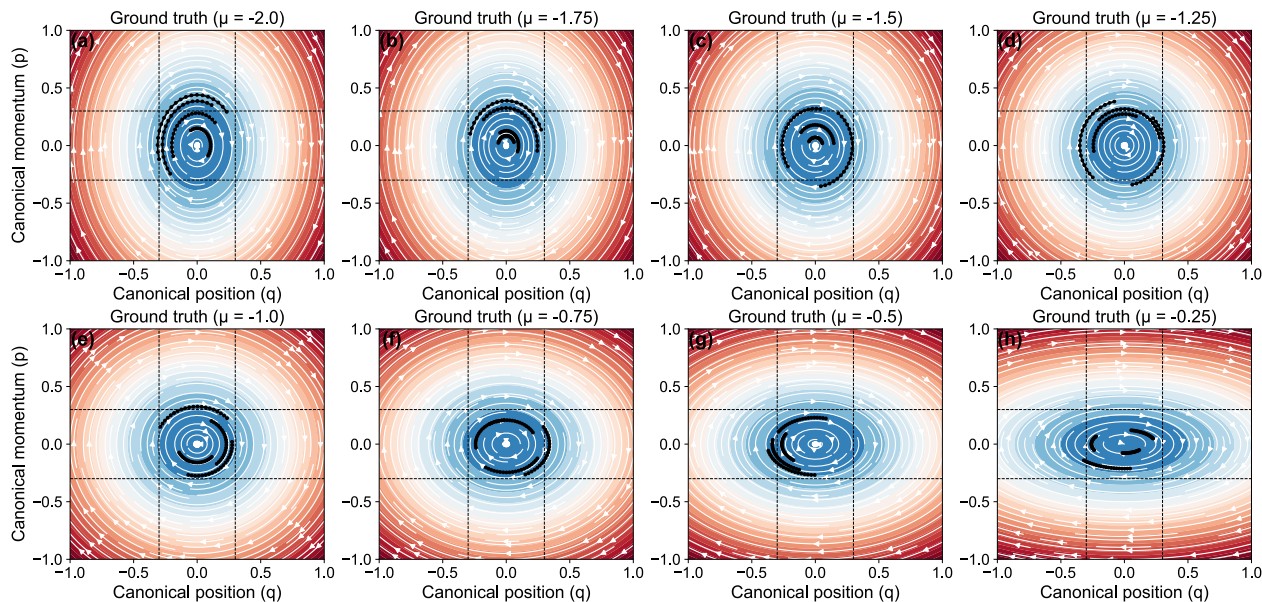

*Figure 18.* Examples of training trajectories from the experiment described in Appendix G.

**Training details.** We basically followed the settings outlined in the original CoDA paper (Kirchmeyer et al., 2022). We trained the context-informed NODEs using the Adam optimizer with default settings for 10,000 epochs, using a learning rate of $10^{-3}$ and a full-batch size. To improve training stability, we employed exponential scheduled sampling for teacher forcing, starting with an initial probability of 0.99 and applying a decay rate of 0.99 every 30 epochs. $\lambda_\xi = 10^{-4}$ and $\lambda_\Omega = 10^{-6}$ were used for the sparsity regularizer $\mathcal{R}(W, \xi_e)$ in (2).

## F. Experiment Details: Section 4.3

**Data preparation.** We trained the context-informed NODEs (2) and their Hamiltonian version (Greydanus et al., 2019) on the Hamiltonian system described in Remark B.2 considering both pre- and post-bifurcation regimes within a restricted training domain. Specifically, we randomly sampled four initial conditions from the uniform distribution $(q(0), p(0)) \sim \mathcal{U}([-0.3, 0.3] \times [-0.3, 0.3])$ for each of the eight parameter values $\mu_e^{\mathrm{tr}} \in \{-0.35, -0.25, -0.15, -0.05, 0.05, 0.15, 0.25, 0.35\}$. For each sampled initial condition, we simulated the dynamics with a time horizon $T = 2.0$ and a time step $\Delta t = 0.1$. This results in $|D_e| = 4$ trajectories per value of $\mu_e^{\mathrm{tr}}$, yielding a total of $|D_e| \times 8 = 32$ training trajectories. The phase portraits of the training trajectories are visualized in Figure 17.

**Architecture.** Note that for the Hamiltonian context-informed NODEs, a neural scalar Hamiltonian function $\mathcal{H}$ is parameterized in the context-informed sense:

$$\dot{\mathbf{x}} = (\dot{q}, \dot{p}) = f(\mathbf{x}, \theta_e = \theta_c + W\xi_e) = (\partial_p \mathcal{H}(\mathbf{x}; \theta_c + W\xi_e), -\partial_q \mathcal{H}(\mathbf{x}; \theta_c + W\xi_e)).$$

Then, this model is trained using (2) in the same manner as described in Section 2. We employed 4-layer MLPs with hidden layers of width 64 and tanh activation functions. We used the fourth-order RK method as an ODE solver.

**Training details.** We basically followed the settings outlined in the original CoDA paper (Kirchmeyer et al., 2022). We trained the context-informed NODEs using the Adam optimizer with default settings for 10,000 epochs, using a learning rate of $10^{-3}$ and a full-batch size. To improve training stability, we employed exponential scheduled sampling for teacher forcing, starting with an initial probability of 0.99 and applying a decay rate of 0.99 every 30 epochs. $\lambda_\xi = 10^{-4}$ and $\lambda_\Omega = 10^{-6}$ were used for the sparsity regularizer $\mathcal{R}(W, \xi_e)$ in (2).

## G. Linear System Identification with the Standard Pre-Bifurcation Training Setting

**Data preparation.** We trained the context-informed NODEs (2) and their Hamiltonian version (Greydanus et al., 2019) on the Hamiltonian system described in Remark B.2 under the pre-bifurcation regime with a restricted training domain.

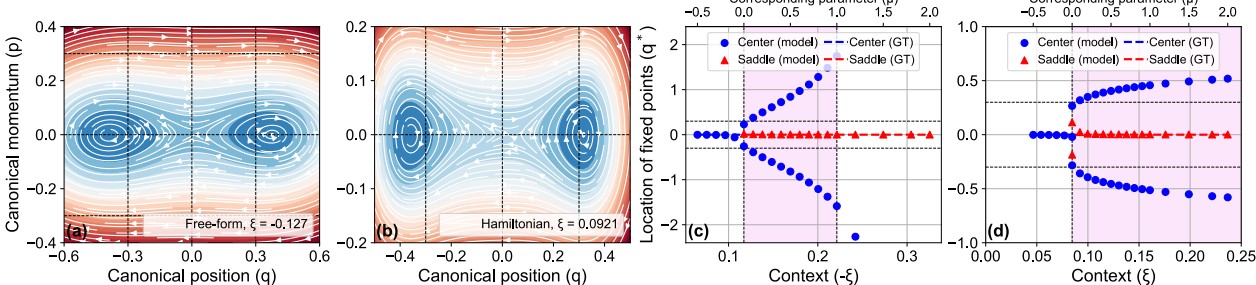

*Figure 19.* Phase portraits of (a) free-form and (b) Hamiltonian models trained with $(\dot{q}, \dot{p}) = (p, \mu q)$, and constructed in a zero-shot manner by linearly extrapolating $\xi_e$, correspond to $\mu_e = 0.1$. Bifurcation diagrams of (c) free-form and (d) Hamiltonian models.

Specifically, we randomly sampled four initial conditions from the uniform distribution $(q(0), p(0)) \sim \mathcal{U}([-0.3, 0.3] \times [-0.3, 0.3])$ for each of the eight parameter values $\mu_e^{\text{tr}} \in \{-2.0, -1.75, -1.5, -1.25, -1.0, -0.75, -0.5, -0.25\}$. For each sampled initial condition, we simulated the dynamics with a time horizon $T = 2.0$ and a time step $\Delta t = 0.1$. This results in $|D_e| = 4$ trajectories per value of $\mu_e^{\text{tr}}$, yielding a total of $|D_e| \times 8 = 32$ training trajectories. It is important to note that all training data consists solely of single-orbit trajectories from the pre-bifurcation symmetric regime. The phase portraits of the training trajectories are visualized in Figure 18.

**Architecture.** We employed 4-layer MLPs with hidden layers of width 64 and `tanh` activation functions, following Appendix F. We used the fourth-order RK method as an ODE solver.

**Training details.** We basically followed the settings outlined in the original CoDA paper (Kirchmeyer et al., 2022). We trained the context-informed NODEs using the Adam optimizer with default settings for 10,000 epochs, using a learning rate of $10^{-3}$ and a full-batch size. To improve training stability, we employed exponential scheduled sampling for teacher forcing, starting with an initial probability of 0.99 and applying a decay rate of 0.99 every 30 epochs. $\lambda_\xi = 10^{-4}$ and $\lambda_\Omega = 10^{-6}$ were used for the sparsity regularizer $\mathcal{R}(W, \xi_e)$ in (2).

**Hallucinated broken symmetry.** Figure 19 (a–b) illustrates the phase portraits explored by context-informed models trained on the linear system under the setting described in Appendix G. As shown, both the free-form and Hamiltonian NODEs misinterpret the bifurcation, incorrectly identifying it as a symmetry-breaking transition, resulting in a spurious double-well structure. The double-well structure in the free-form model collapses with a moderately large $\xi$, reverting to a typical saddle point (Figure 19 (c)). The Hamiltonian model retains the misidentified double-well structure persistently (Figure 19 (d)). In addition, interestingly, the Hamiltonian model locally exhibits a triple-well structure at $\xi = \xi_{\text{crit}}$ (that corresponds to $\mu_e \sim 0$, see Figure 20), though it collapses very rapidly. It can be understood in a similar way that, if the model locally generates two saddle points incorrectly, with their Poincaré indices summing to $-2$, while $(0, 0)$ remains a center with an index of $+1$ and the outer orbit structure is preserved, the model must generate two additional centers

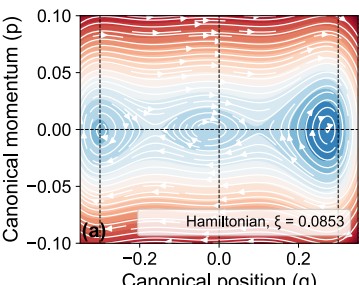

*Figure 20.* The triple-well phase portrait of the Hamiltonian NODE model, correspond to $\mu_e = 0.01$.

with a total index of $+2$ to offset the discrepancy, ensuring that the overall index sums to $-2 + 1 + 2 = +1$ on the outer orbit. However, as the model correctly transitions the center at $(0, 0)$ into a saddle point, this structure quickly collapses, giving way to a (still spurious) double-well structure to maintain a total index of $+1$.

## H. Experiment Details: Section 5.1

**Data preparation.** We trained the context-informed NODEs (2) on the cusp bifurcating system under the pre-bifurcation regime exclusively. Specifically, we randomly sampled four initial conditions from the uniform distribution $(q(0), p(0)) \sim \mathcal{U}([-2.0, 2.0] \times [-2.0, 2.0])$ for each of the 16 parameter vectors $(\mu_e^{\text{tr}}, \nu_e^{\text{tr}}) \in \{-2.0, -1.5, -1.0, -0.5\}^2$. For each sampled initial condition, we simulated the dynamics with a time horizon $T = 2.0$ and a time step $\Delta t = 0.1$. This results in $|D_e| = 4$ trajectories per vector of $(\mu_e^{\text{tr}}, \nu_e^{\text{tr}})$, yielding a total of $|D_e| \times 16 = 64$ training trajectories. It is important to note that all training data consists solely of monostable trajectories from the pre-bifurcation regime.

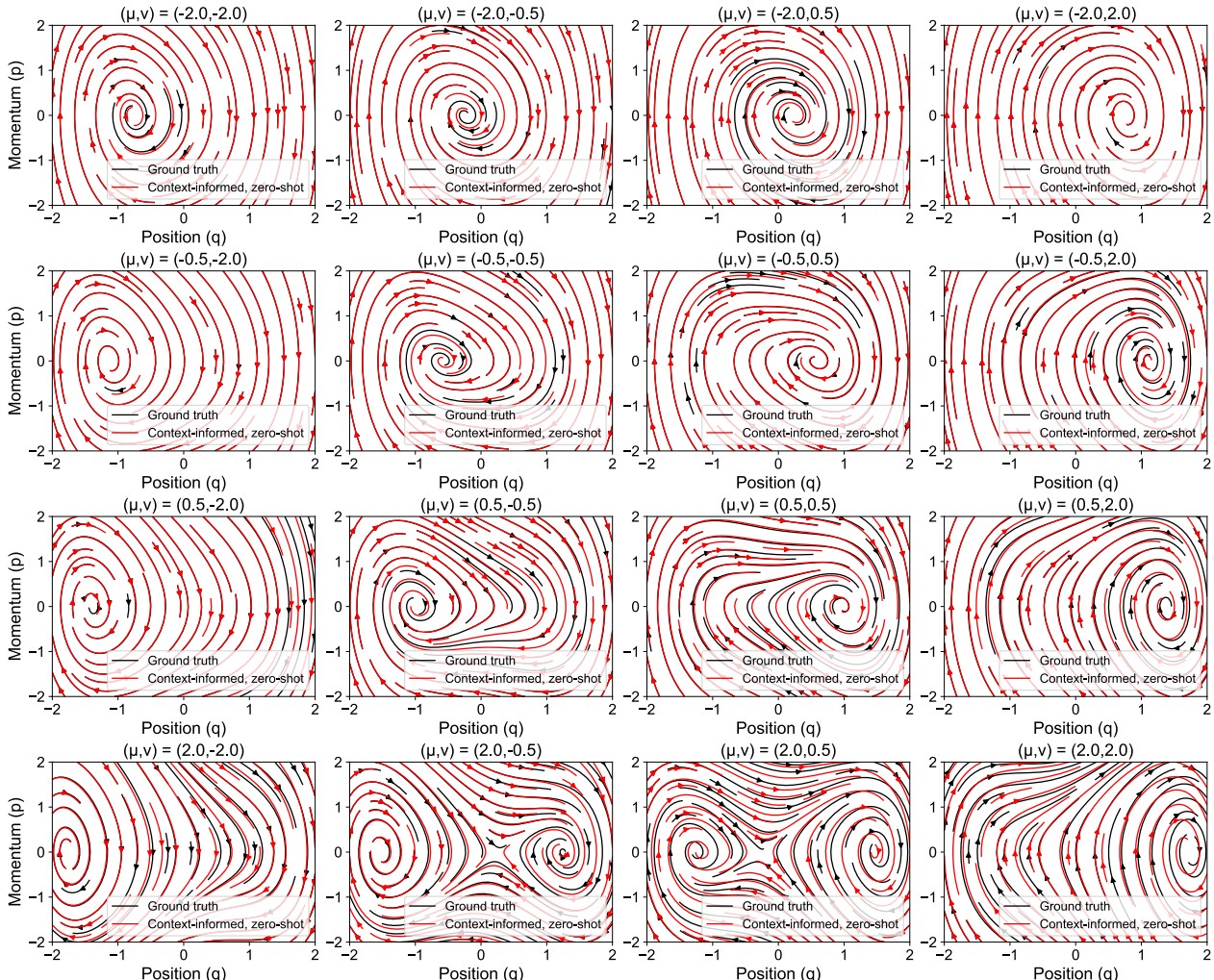

*Figure 21.* Phase portraits constructed in a zero-shot manner by linearly extrapolating the context vector $\xi_e$ based on pre-bifurcation training data, corresponding to different values of $(\mu_e, \nu_e)$.

**Architecture.** We basically followed the settings outlined in the original CoDA paper (Kirchmeyer et al., 2022). We employed 4-layer MLPs with hidden layers of width 64 and `swish` activation functions. We used the fourth-order RK method as an ODE solver.

**Training details.** We basically followed the settings outlined in the original CoDA paper (Kirchmeyer et al., 2022). We trained the context-informed NODEs using the Adam optimizer with default settings for 50,000 epochs, using a learning rate of $10^{-3}$ and a full-batch size. To improve training stability, we employed exponential scheduled sampling for teacher forcing, starting with an initial probability of 0.99 and applying a decay rate of 0.99 every 30 epochs. $\lambda_\xi = 10^{-4}$ and $\lambda_\Omega = 10^{-6}$ were used for the sparsity regularizer $\mathcal{R}(W, \xi_e)$ in (2).

**Evaluation details.** After training the model, we constructed a mesh grid over $(\mu_e^{\text{test}}, \nu_e^{\text{test}}) \in [-2.0, 2.0]^2$ with intervals of $\Delta\mu_e^{\text{test}} = 0.1$ and $\Delta\beta_e^{\text{test}} = 0.1$, resulting in a total of $41 \times 41 = 1681$ parameter combinations. For each parameter vector $(\mu_e^{\text{test}}, \nu_e^{\text{test}})$, the vector field of the learned context-informed NODE model was obtained by inputting the corresponding context vector $\xi_e^{\text{test}}$, which was estimated using a linear regression model fitted on the relationship between the training parameters $(\mu_e^{\text{tr}}, \nu_e^{\text{tr}})$ and their associated context vectors $\xi_e^{\text{tr}}$. Then, the bifurcation surface shown in Figure 10 (a) was generated by identifying the fixed points of the modeled vector field at each $\xi_e^{\text{test}}$, using the neural-adjoint method described in Appendix C. Figure 10 (b) directly compares the ground truth and the modeled vector fields for some selected parameter vectors near the catastrophic transition. In addition, an extended comparison over a broader range of parameters $(\mu_e, \nu_e)$ is presented in Figure 21.

# I. Poincaré–Hopf Regularization

The Poincaré–Hopf regularization is formulated to minimize the following topological discrepancy by guiding the model's Poincaré indices toward the desired values:

$$\mathcal{R}_{\mathrm{PH}}(\theta_c, W, \xi) = \sum_i \mathbb{E}_{\Gamma \sim p_\Gamma(\Gamma | \chi_i)} \left[ D(\mathrm{Ind}(f(\cdot; \theta_c + W\xi), \Gamma) \| \chi_i) \right],$$

where $D(\cdot \| \cdot)$ represents a distance function, $\chi_i$ is the $i$-th desired index value, $\xi$ is the context vector, and $p_\Gamma$ is a distribution over the test contours $\Gamma$. Because the appropriate test contour depends on the desired index, $p_\Gamma$ is conditioned on $\chi$ (e.g., $+1$ for a global test contour or $-1$ for a local test contour in the case of double-well potentials). The model's index $\mathrm{Ind}(f, \Gamma)$ is computed by numerically evaluating the two-dimensional contour integral described in Remark 4.1:

$$\mathrm{Ind}(f, \Gamma) = \frac{1}{2\pi} \sum_{i=0}^{M-1} \frac{-f_{p,i} \Delta f_{q,i} + f_{q,i} \Delta f_{p,i}}{\|f_i\|^2 + \epsilon},$$

where $M$ is the number of segments used to discretize $\Gamma$ and $f_i = (f_{q,i}, f_{p,i})^T$ denotes the vector field at the $i$-th segment of $\Gamma$. The terms $\Delta f_{q(p),i} = f_{q(p),(i+1) \bmod M} - f_{q(p),i}$ represent the finite differences of the field components. The small constant $\epsilon = 10^{-4}$ is added to avoid division by zero when $\|f_i\| \to 0$.

In the experiment described in Section 5.2 (and detailed in Appendix J), we used a simplified version of the regularizer by fixing $\xi$ to $\xi_e, e \in \mathcal{E}_{\mathrm{tr}}$ and setting $\Gamma$ to predetermined global and local test contours as follows:

$$\mathcal{R}_{\mathrm{PH}}(\theta_c, W, \xi_e) = \lambda_{\mathrm{PH}} \left[ \mathcal{R}_{\mathrm{PH}}^g(\theta_c, W, \xi_e) + \mathcal{R}_{\mathrm{PH}}^l(\theta_c, W, \xi_e) \right], \tag{7}$$

where $\mathcal{R}_{\mathrm{PH}}^g(\theta_c, W, \xi_e)$ and $\mathcal{R}_{\mathrm{PH}}^l(\theta_c, W, \xi_e)$ respectively measure the global and local index mismatches, defined as

$$\mathcal{R}_{\mathrm{PH}}^g = \frac{1}{|\mathcal{E}_{\mathrm{tr}}|} \sum_{e \in \mathcal{E}_{\mathrm{tr}}} \|\mathrm{Ind}(f(\cdot; \theta_c + W\xi_e), \Gamma_{\mathrm{PH}}^g) - 1\|_2^2,$$

$$\mathcal{R}_{\mathrm{PH}}^l = \frac{1}{|\{e | \alpha < 0\}|} \sum_{e | \alpha < 0} \|\mathrm{Ind}(f(\cdot; \theta_c + W\xi_e), \Gamma_{\mathrm{PH}}^l) + 1\|_2^2 + \frac{1}{|\{e | \alpha > 0\}|} \sum_{e | \alpha > 0} \|\mathrm{Ind}(f(\cdot; \theta_c + W\xi_e), \Gamma_{\mathrm{PH}}^l) - 1\|_2^2.$$

Here, $\Gamma_{\mathrm{PH}}^g$ represents a global contour centered at $(0, 0)$, with a major axis length of $r_\psi = 4.0$ and a minor axis length of $r_{\psi'} = 0.5$. Meanwhile, $\Gamma_{\mathrm{PH}}^l$ defines a local contour as a circle with a radius of 0.5. Note that the desired index for the global regularization $\mathcal{R}_{\mathrm{PH}}^g$ is fixed at $+1$, following the Poincaré–Hopf theorem. For the local regularization $\mathcal{R}_{\mathrm{PH}}^l$, the desired index is set to $+1$ for $\alpha > 0$ (i.e., the single- and triple-well cases) and to $-1$ for $\alpha < 0$ (i.e., the double-well dynamics), which requires more specific prior knowledge. In Appendix K, we compare the full regularizer defined in (7), i.e., $\mathcal{R}_{\mathrm{PH}} = \lambda_{\mathrm{PH}}[\mathcal{R}_{\mathrm{PH}}^g + \mathcal{R}_{\mathrm{PH}}^l]$, with its global-only variant $\mathcal{R}_{\mathrm{PH}} = \lambda_{\mathrm{PH}} \mathcal{R}_{\mathrm{PH}}^g$. The hyperparameter $\lambda_{\mathrm{PH}}$ is set to $10^{-3}$.

# J. Experiment Details: Section 5.2

**Data preparation.** We trained the conventional context-informed NODEs and their topologically regularized counterpart with (7) on the Landau–Khalatnikov (LK) system (4). For training, we randomly sampled 4 initial conditions from the uniform distribution $(\psi(0), \psi'(0)) \sim \mathcal{U}([-2.0, 2.0] \times [-2.0, 2.0])$ for each combination of $\alpha_e^{\mathrm{tr}} \in \{-0.4, -0.2, 0.2, 0.4\}$ and $\beta_e^{\mathrm{tr}} \in \{-0.4, -0.2, 0.2, 0.4\}$, resulting in 16 different parameter vectors. To ensure the physical relevance of (4), $\gamma = 0.05$ and $\kappa = 0.5$ were fixed. For each sampled initial condition, we simulated the dynamics over $T = 2.0$ with $\Delta t = 0.1$. This results in $|D_e| = 4$ trajectories per value of $\mu_e^{\mathrm{tr}}$, yielding a total of $|D_e| \times 16 = 64$ training trajectories.

**Architecture.** We basically followed the settings outlined in the original CoDA paper (Kirchmeyer et al., 2022). We employed 4-layer MLPs with hidden layers of width 64 and `swish` activation functions. We used the fourth-order RK method as an ODE solver.

**Training details.** We basically followed the settings outlined in the original CoDA paper (Kirchmeyer et al., 2022). We trained the context-informed NODEs using the Adam optimizer with default settings for 50,000 epochs, using a learning rate of $10^{-3}$ and a full-batch size. To improve training stability, we employed exponential scheduled sampling for teacher forcing, starting with an initial probability of 0.99 and applying a decay rate of 0.99 every 30 epochs. $\lambda_\xi = 10^{-4}$ and $\lambda_\Omega = 10^{-6}$ were used for the sparsity regularizer $\mathcal{R}(W, \xi_e)$ in (2). The regularizer (7) described in Appendix I is added to (2) for the topologically regularized models. We repeated the experiment five times using different random initializations.

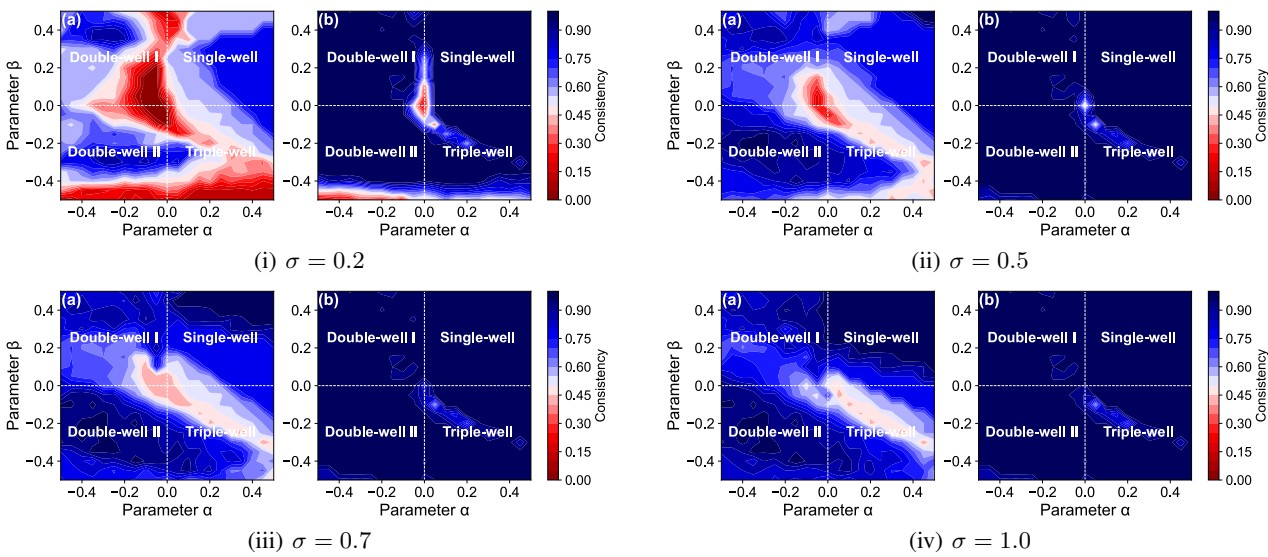

(i) $\sigma = 0.2$  (ii) $\sigma = 0.5$

(iii) $\sigma = 0.7$  (iv) $\sigma = 1.0$

*Figure 22.* Contour plots of the mean consistency scores for (a) the conventional model and (b) the fully regularized model using (7).

**Evaluation details.** After training the models, we evaluated the models' ability to capture the long-term behavior and phase topology of the dynamical system by computing the $L_2$ distance between the limit sets of the ground truth and those generated by each model. We constructed a mesh grid over $(\alpha_e^{\text{test}}, \beta_e^{\text{test}}) \in [-0.5, 0.5]^2$ with intervals of $\Delta\alpha_e^{\text{test}} = 0.05$ and $\Delta\beta_e^{\text{test}} = 0.05$, resulting in a total of $21 \times 21 = 441$ parameter combinations. For each parameter vector $(\alpha_e^{\text{test}}, \beta_e^{\text{test}})$, we randomly sampled $N = 32$ initial conditions from the uniform distribution $(\psi(0), \psi'(0)) \sim \mathcal{U}([-2.0, 2.0] \times [-2.0, 2.0])$ and simulated long-term trajectories with $T = 100.0$ and $\Delta t = 0.1$. Context-informed NODE models and their regularized counterparts were simulated under the same $T$ and $\Delta t$. These simulations utilized 441 context vectors, constructed using linear regression fitted on the relationships between the training parameters $(\alpha_e^{\text{tr}}, \beta_e^{\text{tr}})$ and the corresponding context vectors $\xi_e^{\text{tr}}$. The limit set for each dynamics model is defined as the converged attractors $(\psi^*, \psi'^*) = \mathbf{x}^* \simeq \mathbf{x}(T)$. We empirically found that $T = 100.0$ is sufficient to approximate the limit sets accurately. After obtaining $\mathbf{x}(T)$ for both the ground truth and the tested NODE models, the topological consistency score of $e$-th test parameter vector $(\alpha_e^{\text{tr}}, \beta_e^{\text{tr}})$ is calculated as $s_e = \frac{1}{N} \sum_{i=1}^{N} \mathbb{1}(\|\mathbf{x}_e^i(T) - \hat{\mathbf{x}}_e^i(T)\|_2 < \sigma)$, where $\sigma$ is a threshold distance, and $\mathbf{x}_e^i(T)$ and $\hat{\mathbf{x}}_e^i(T)$ represent the state vectors at $T$ of the ground truth dynamics and the NODE models, respectively, starting from the $i$-th test initial condition for the $e$-th parameter vector.

Figure 22 presents contour plots of the topological consistency scores as the threshold $\sigma$ varies. As shown, the consistency scores of conventional context-informed NODE models show a limited improvement despite increasing $\sigma$, particularly in the triple-well regions ($\alpha > 0, \beta < 0$). This suggests that the vanilla context-informed NODE struggles to capture the correct topology, leading to diverging behavior. On the other hand, the topologically regularized model demonstrates enhanced long-term predictability. We also report the Mean Absolute Percentage Error (MAPE), which provides a more reliable basis for performance comparison across different parameters than MSE (Kirchmeyer et al., 2022), as shown in Figure 23. This

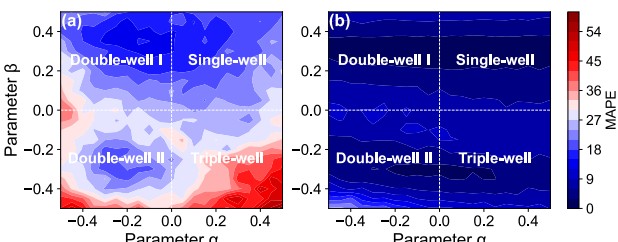

*Figure 23.* Contour plots of the MAPEs for (a) the conventional model and (b) the fully regularized model using (7).

further validates the effectiveness of the proposed Poincaré–Hopf regularization.

## K. Ablation Study of Poincaré–Hopf Regularization

**Topological regularization without a local prior.** The proposed regularization (7) requires prior knowledge at both global ($\mathcal{R}_{\text{PH}}^g$) and local ($\mathcal{R}_{\text{PH}}^l$) levels. Local regularization typically demands more detailed, system-specific information. For example, computing $\mathcal{R}_{\text{PH}}^l$ in (7) necessitates prior knowledge about the transition behavior of the local index from $-1$

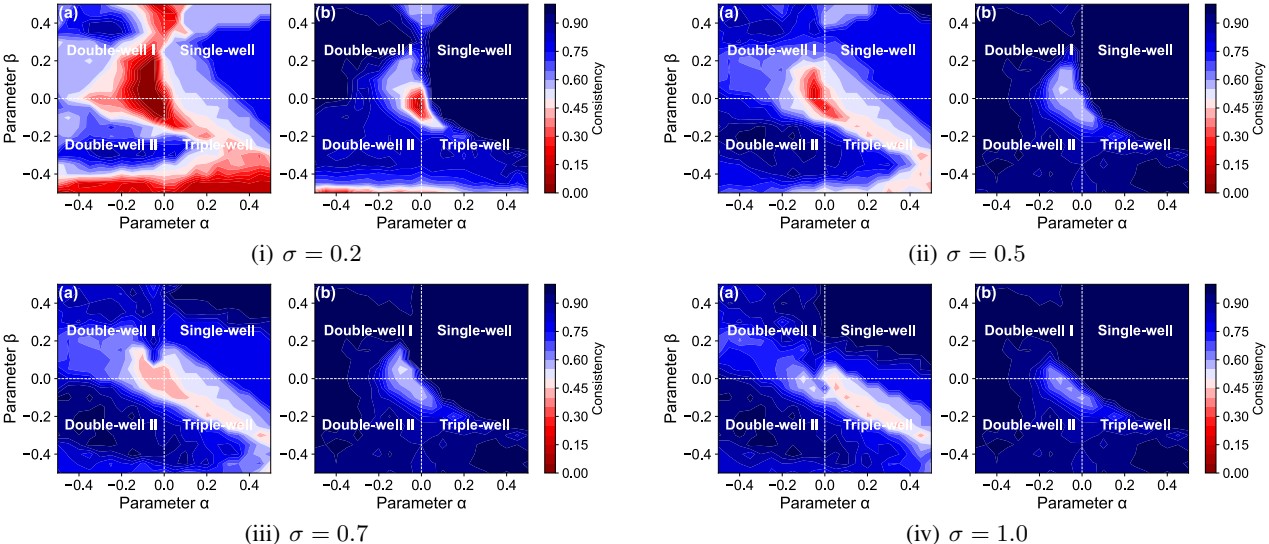

*Figure 24.* Contour plots of the mean consistency scores for (a) the conventional context-informed NODE and (b) the regularized version with only the global constraint.

to +1 as a function of $\alpha$. In contrast, applying only global constraints requires significantly less prior information. As stated in Theorem 4.1, the global regularization term $\mathcal{R}^g_{\mathrm{PH}}$ relies only on coarse-grained knowledge about the phase manifold where the ODE is defined; for instance, in the LK experiment, $\mathcal{R}^g_{\mathrm{PH}}$ only requires that the global index constraint equals +1, regardless of parameters. Despite its simplicity, this type of global information can be especially valuable when detailed prior knowledge about the target system is limited.

To evaluate the effectiveness of topological regularization under such minimal assumptions, we revisited the LK experiment discussed in Appendix J using only the global term, i.e., $\mathcal{R}_{\mathrm{PH}}(\theta_c, W, \xi_e) = \lambda_{\mathrm{PH}}\mathcal{R}^g_{\mathrm{PH}}(\theta_c, W, \xi_e)$. All other experimental settings remained unchanged. Figure 24 compares the mean consistency scores between the vanilla model and the globally regularized model across different values of $\sigma$, following the evaluation protocol described earlier. The results show that the globally regularized model (Figure 24 (b)) significantly outperforms the vanilla model (Figure 24 (a), identical to Figure 22

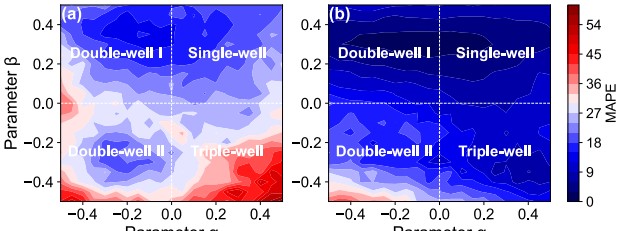

*Figure 25.* Contour plots of the MAPEs for (a) the conventional model and (b) the regularized version with only the global constraint.

(a)), even when relying solely on global information. While its performance is slightly lower than that of the fully regularized model with both local and global constraints (Figure 22 (b)), this reflects a trade-off between the cost of incorporating prior knowledge and achieving optimal performance. We also report the MAPE profiles in Figure 25, which lead to similar results. These findings suggest that imposing global topological constraints via the Poincaré–Hopf theorem provides a cost-effective and robust form of topology-aware regularization, particularly in settings with limited prior knowledge.

**Training Dynamics Analysis.** The proposed Poincaré–Hopf regularization described in Appendix I is based on a finite difference method, resulting in computational overhead proportional to the number of discretized points $M$. In all our experiments, we discretized the test contours using $M = 128$ points, which led to a moderate increase in runtime under identical computational environments (vanilla: $52.9 \pm 5.5$ ms/epoch; global-only regularized: $65.7 \pm 5.1$ ms/epoch; fully regularized: $77.6 \pm 4.7$ ms/epoch). Despite this additional cost, the regularized model demonstrates faster convergence in practice and significantly improves test accuracy, thanks to the guidance provided by topological regularization.

Figure 26 compares the vanilla, globally regularized, and fully regularized models in terms of the following metrics, all plotted over normalized wall-clock time: (a) the global regularization value $\mathcal{R}^g_{\mathrm{PH}}$; (b) the local regularization value $\mathcal{R}^l_{\mathrm{PH}}$; (c) the test MSE on a linear time axis, and (d) the test MSE on a logarithmic time axis. As defined in (7), the regularization values directly reflect index mismatches between the model and the ground truth, making them useful for evaluating whether

*Figure 26.* Comparison between the vanilla and regularized models in terms of (a) the global regularization value $\mathcal{R}_{\mathrm{PH}}^g$, (b) the local regularization value $\mathcal{R}_{\mathrm{PH}}^l$, (c) the test MSE on a linear time axis, and (d) the test MSE on a logarithmic time axis. In (a) and (b), the black dashed line indicates the discretization error limit corresponding to the ground truth dynamics.

the model adheres to the ground truth topology. Note that while all models compute both $\mathcal{R}_{\mathrm{PH}}^g$ and $\mathcal{R}_{\mathrm{PH}}^l$, only the fully regularized model is explicitly trained to minimize both terms. The vanilla model does not use any form of regularization, whereas the globally regularized model is constrained solely by the global term $\mathcal{R}_{\mathrm{PH}}^g$. The test MSE was computed in the same way as the training MSE loss, but using 32 trajectories per parameter that were different from those used during training. All of these loss profiles are from the previously discussed LK experiment.

As shown in Figure 26 (a), both the globally and fully regularized models maintain a low and stable global index mismatch, converging to $4 \times 10^{-3}$ with minimal variance, as they jointly minimize data loss and the global topological loss. It is worth noting that even the ground truth vector field incurs a nonzero global topological loss of $4 \times 10^{-3}$ due to discretization error introduced by the finite approximation of the contour integral. In contrast, the vanilla model, which lacks any topological constraints, exhibits a significantly higher global topological error exceeding $10^{-2}$, along with substantial variance. These results suggest that the vanilla model fails to reliably capture the correct global topological structure, highlighting the importance of incorporating topological regularization.

In Figure 26 (b), the fully regularized model, which explicitly minimizes the local regularization term, exhibits a sharp decline in this value and maintains it at a consistently low level of $2 \times 10^{-4}$, which corresponds to the discretization limit of the ground truth. Meanwhile, the vanilla model shows a gradual decrease, as it implicitly captures certain topological features from the data, but its learning efficiency is significantly lower than that of the regularized model. Interestingly, the global-only model, despite lacking the local regularizer, also achieves a noticeable and consistent reduction in the local index discrepancy, though less effectively than the fully regularized model. This phenomenon can be attributed to the discussion in Section 4: the global topological constraint—the total Poincaré index of $+1$—naturally encourages the model to learn the correct local topological structure that governs bifurcation, in accordance with the Poincaré–Hopf theorem.

This difference in topological learning leads to faster convergence and better predictive accuracy in the regularized cases, as illustrated in Figure 26 (c–d). Between the globally and fully regularized models, the latter demonstrates superior performance, consistent with the earlier discussion of the trade-off between incorporating prior knowledge and achieving optimal performance. Overall, these findings underscore the effectiveness and efficiency of topology-aware regularization.

