# OpenReview forum: "Context-Informed Neural ODEs Unexpectedly Identify Broken Symmetries: Insights from the Poincaré–Hopf Theorem"
_ICML.cc/2025/Conference — ICML 2025 poster_

### Official Review · Reviewer_xWSr · 2025-03-09

**Overall Recommendation:** 3

**Summary:**

The paper introduces Context-Informed Neural ODEs (CI-NODEs), a framework designed to learn dynamical systems exhibiting bifurcation behaviors, particularly symmetry-breaking bifurcations. The authors claim that CI-NODEs, despite being trained solely on pre-bifurcation, symmetric data, can predict post-bifurcation symmetry-breaking behaviors without explicit physics-based priors. They attribute this phenomenon to the implicit use of topological invariants, specifically the Poincaré index, and provide a formal explanation via the Poincaré-Hopf theorem. Additionally, a novel topological regularization term inspired by this theorem is proposed and tested on the Landau-Khalatnikov system to enhance generalization.

**Claims And Evidence:**

The claim that CI-NODEs can "identify" bifurcations and predict post-bifurcation behaviors is intriguing but lacks rigorous justification. The results indicate that the model sometimes hallucinates bifurcations, which contradicts the assertion that the approach reliably generalizes.

The use of the Poincaré-Hopf theorem to explain the model’s behavior is an interesting theoretical contribution, but the practical implications remain unclear. There is no empirical validation showing that the theorem correctly predicts when the model will succeed or fail.

The claim that the proposed regularization enhances generalization is weakly supported. The experiments show some improvements, but they lack robustness tests, such as evaluations across a broader range of bifurcation scenarios.

**Essential References Not Discussed:**

None.

**Experimental Designs Or Analyses:**

The experiments provide some compelling qualitative results, but the lack of quantitative robustness testing is a significant weakness.

The comparisons with baseline methods, such as traditional NODEs, are somewhat superficial. More rigorous benchmarking against established bifurcation detection techniques would strengthen the claims.

The zero-shot generalization experiments are interesting but suffer from a lack of statistical analysis. How often does the model correctly infer post-bifurcation behavior versus hallucinating incorrect structures?

**Methods And Evaluation Criteria:**

The experimental setup is well-structured but suffers from narrow validation. The datasets used for training and testing are limited to specific types of bifurcations, making it unclear how well the model generalizes to other dynamical systems.

The chosen evaluation metrics (e.g., Mean Squared Error, trajectory consistency) are appropriate but insufficient to fully assess model reliability. Additional evaluations such as robustness to noise, long-term stability, and sensitivity to initial conditions would be beneficial.

The ablation studies focus mainly on the impact of context-informed modeling but do not adequately assess individual architectural choices, such as the specific design of the fusion mechanism.

**Other Comments Or Suggestions:**

(1) Provide statistical significance testing for key experimental results.

(2) Clarify how computational complexity scales with increasing problem size and longer time horizons.

(3) Conduct additional experiments to test the model’s robustness to noise and distribution shifts.

(4) Improve clarity by restructuring the theoretical discussion to make the key insights more accessible.

**Other Strengths And Weaknesses:**

**Strengths**:

The paper presents a novel perspective on learning bifurcating dynamical systems using NODEs.

The use of topological invariants as an implicit learning signal is an interesting theoretical contribution.

The proposed topological regularization is conceptually novel and may inspire further research in constrained learning for dynamical systems.

**Weaknesses**:

The empirical validation is not rigorous enough to support the paper’s ambitious claims.

The theoretical arguments, while insightful, are largely heuristic and lack formal proof.

The experimental design does not adequately explore failure cases, making it difficult to assess the model’s reliability.

The paper is dense and difficult to follow, with crucial details buried in the appendix.

**Questions For Authors:**

(1) Can you provide more evidence that CI-NODEs do not hallucinate bifurcations in cases where no symmetry breaking occurs?

(2) How does CI-NODE compare to physics-informed methods, such as PINNs, in terms of accuracy and interpretability?

(3) What guarantees, if any, can be provided regarding the reliability of CI-NODEs in predicting post-bifurcation behaviors?

(4) Have you tested CI-NODEs on higher-dimensional bifurcation problems, and if so, how does it perform?

(5) How does the choice of context representation affect the model’s ability to generalize?

**Relation To Broader Scientific Literature:**

The claim that this work offers new insights into OOD generalization in dynamical systems is overstated. While the results are interesting, they do not establish a broadly applicable principle for OOD learning.

**Theoretical Claims:**

The theoretical contribution is a highlight of the paper but remains somewhat speculative. While the Poincaré-Hopf theorem provides an insightful perspective, the connection between the theorem and the model’s emergent properties is not formally established.

There is no rigorous proof explaining why CI-NODEs should be able to generalize beyond training data in the specific manner observed. The argument based on topological constraints is heuristic at best.

The discussion of the relationship between NODEs and dynamical systems theory is useful but lacks depth in explaining how these theoretical insights could be leveraged to improve practical performance.

---

> ### Author Rebuttal · Authors · 2025-03-31
>
> We sincerely appreciate the reviewer’s thoughtful and constructive feedback. Below, we address each comment carefully. Additional experimental results are available in the **README** at https://github.com/anonymous-account123/icml2025-7637 and will be thoroughly incorporated into our revised paper.
> ***
> **Q1. Reliability of the zero-shot identification.**
>
> We fully agree with your comment that evaluating the reliability of bifurcation identification is of critical importance. To statistically validate the robustness of zero-shot bifurcation identification, we conducted repeated experiments (five runs) using different random seeds and initial conditions. Consistently, the model reliably identified post-bifurcation behaviors, demonstrating statistical robustness and reproducibility (**Figure 1(a)** at the provided link).
>
> In addition, we performed additional robustness tests under realistic scenarios, including noisy observations and limited training data. These challenging settings revealed that the model remains capable of accurately identifying symmetry-breaking bifurcations, albeit with slightly increased variance compared to the ideal noise-free scenario (**Figure 1(b-c)** at the provided link).
> ***
> **Q2. Hallucinated bifurcations and empirical validation of the theorem.**
>
> The hallucinated symmetry-breaking scenario in Section 4.3 is purposefully designed to illustrate the predictive capability of Proposition 4.1. In the analyzed linear system, a simple center-to-saddle bifurcation is expected if the model merely approximates the functional form. However, our empirical results demonstrate a spurious symmetry-breaking bifurcation, as predicted by Proposition 4.1, confirming that the model genuinely leverages topological constraints rather than simple functional approximation. We will clarify this critical interpretation in the revised paper as you suggested.
> ***
> **Q3. How to detect and avoid hallucinated bifurcations.**
>
> We completely agree that identifying whether a model hallucinates bifurcations is vital. We propose a straightforward yet robust criterion: evaluating the variance in bifurcation diagrams generated by multiple independent training runs. Correctly identified bifurcations show minimal variance, indicating stable and consistent identification. Conversely, hallucinated bifurcations yield significantly higher variance due to their structural instability. **Figure 2** at the provided link clearly demonstrates this differentiation approach.
> ***
> **Q4. Other type of bifurcations or dynamical systems.**
>
> Following your suggestion, we investigated a non-Hamiltonian dynamical system exhibiting a codimension-2 cusp bifurcation: $(\dot{x}, \dot{y}) = (y, b + a x - x^3 – d y)$, with variable parameters $(a, b)$ and fixed $d = 0.5$. This system serves as a canonical model for capturing catastrophic transitions and hysteresis phenomena, which are fundamental in science and engineering. Remarkably, despite being trained only on the simple, monostable pre-bifurcation regime $(a,b) \in \[-2.0, -1.5, -1.0, -0.5 ]^2$, the model successfully identified the cusp bifurcation surface, validating the model's broader applicability. Detailed visualizations are available in **Figure 3** at the provided link.
> ***
> **Q5. Comparison with PINNs.**
>
> We acknowledge that PINNs effectively utilize explicit physical priors. However, they become inapplicable if physical laws are unknown or incorrectly assumed. Our proposed approach circumvents this limitation by leveraging general topological invariants. For instance, models representing vector fields on a sphere (e.g., climate models) naturally adopt a global index constraint of $\chi(S^2) = +2$. Such universally applicable prior knowledge positions our approach as complementary and potentially advantageous in scenarios where explicit physical laws are unavailable. This point will be discussed in the revised manuscript.
> ***
> **Q6.  Computational complexity.**
>
> Modeling $N$ environments using vanilla NODEs with $M$ parameters requires $N \times M$ parameters. Conversely, the CI-NODEs reduce this requirement to approximately $M(1 + K)$, where $K$ represents the context dimensionality, showing substantial parameter efficiency, particularly when $N > K$.
>
> Regarding computational overhead from topological regularization, our experiments confirm a moderate increase in computation time (from 53.5 ms/epoch for vanilla models to 80.9 ms/epoch for regularized models under the same environment). Nevertheless, the regularized model benefits from improved learning stability and accordingly demonstrates faster empirical convergence, as shown in **Figure 4** at the provided link. We will clearly address this trade-off in the revision.
> ***
> **Q7. Clarity of the paper.**
>
> We appreciate your suggestions for clarity improvement. We will refine the main text and Appendix to highlight essential details clearly, and will include the additional discussions and experiments made during the rebuttal.

---

> > ### Comment · Reviewer_xWSr · 2025-04-03
> >
> > I appreciate the authors' detailed rebuttal. I will increase my score.

---

> > > ### Author Response · Authors · 2025-04-09
> > >
> > > We are pleased that our response has addressed your concerns. We sincerely appreciate your thoughtful suggestions and kind recognition of our work.

---

### Official Review · Reviewer_XQYv · 2025-03-13

**Overall Recommendation:** 4

**Summary:**

The paper finds that context-informed Neural Ordinary Differential Equations (NODEs) can identify symmetry-breaking bifurcations in dynamical systems (DS). By leveraging topological invariants like the Poincaré index and the Poincaré-Hopf theorem,  the paper demonstrates conditions under which context-informed NODEs can out-of-domain (OOD) generalize without explicit physics-based priors. The manuscript further introduces a regularization based on these findings and empirically demonstrate its use on the Landau-Khalatnikov system.

**Claims And Evidence:**

All claims are supported by evidence, even though some more evidence would be helpful (see **Weaknesses**).

**Essential References Not Discussed:**

Not essential per se, but [1] backs the findings in that they also find that a similar hierarchical approach to CoDA, but using an recurrent neural network backbone, leads to linear relationships between GT control parameter and context vectors with capabilities of inter-and-extrapolation.

**Experimental Designs Or Analyses:**

Experimental Analyses seem appropriate.

**Methods And Evaluation Criteria:**

Methods and evaluation criteria are appropriate.

**Other Comments Or Suggestions:**

**Typo**: “This raises a pertinent question arises:” (p. 5, ll. 273-274)

**References**

[1] Brenner, M. et al. (2024). Learning Interpretable Hierarchical Dynamical Systems Models from Time Series Data. (ICLR 2025)

[2] Göring, N.A. et al. (2024). Out-of-Domain Generalization in Dynamical Systems Reconstruction. (ICML 2024)

**Other Strengths And Weaknesses:**

**Strengths**: I think the paper does a great job of motivating, explaining and presenting each and every experiment in a detailed fashion.

**Weaknesses**: A current weakness of the manuscript is that the hypothesized mechanism for the possible OOD generalization, i.e. the Poincaré-Hopf theorem,  is only validated for NODE backbone based methods and tested on Hamiltonian systems. To remedy this weakness, the authors could try different DSR backbones, e.g. RNNs, and apply the regularizer to different, non-Hamiltonian DS.

**Questions For Authors:**

1. The choice of $\chi_{PH}$ (desired index) is a quantity that does depend on physical priors we have about the observed DS, correct? If so, how can the regularizer help in cases where this information is a priori *not* available?
2. I’m struggling a bit with section 4.3; without knowledge of the GT system, is there a way to identify whether the model is hallucinating the wrong bifurcation?

**Relation To Broader Scientific Literature:**

Training on a limited window of ground-truth (GT) control parameter values using a hierarchical/meta-learning approach and extrapolating beyond this range has also been done in a previous study, albeit not across bifurcations [1]. However, similar linear relationships between context vector and GT control parameters have been observed (see Fig. 3a). Also, as mentioned in the current manuscript, [2] showed that *vanilla* DS reconstruction methods (such as vanilla NODEs) struggle to generalize across state space in a multistable (and hence) OOD scenario. In this context, this work is novel as it shows conditions under which OOD generalization is indeed possible.

**Theoretical Claims:**

The paper's theoretical findings are based on established DS theory, such as the Poincaré-Hopf theorem. Appropriate references thereof are provided (e.g. Strogatz 2018). I did not check the proof for Proposition 4.1.

---

> ### Author Rebuttal · Authors · 2025-03-31
>
> We sincerely appreciate the reviewer’s thoughtful and constructive feedback. Below, we address each comment carefully. Additional experimental results are available in the **README** at https://github.com/anonymous-account123/icml2025-7637 and will be thoroughly incorporated into our revised paper.
> ***
> **Q1. Brenner et al.**
>
> Thank you for pointing out the relevant reference, Brenner et al. (2024), which also enables both interpolation and extrapolation of model parameters via linear modulation of subject-specific features. However, as you noted, it does not address the bifurcation-driven OOD, specifically the case where training is conducted solely in the pre-bifurcation regime while the post-bifurcation regime is explored at test time. We will cite this work and include a discussion in the revised paper.
> ***
> **Q2. How to detect and avoid hallucinated bifurcations.**
>
> The hallucinated symmetry-breaking experiment in Section 4.3 is intentionally designed as a pathological case to illustrate how the model mistakenly classifies a simple center-to-saddle bifurcation as a symmetry-breaking one, as predicted by Proposition 4.1. In the linear system considered in Section 4.3, if the NODE model is merely approximating a known functional form, it should undergo a simple center-to-saddle bifurcation. However, the experimental results show that the NODE model instead undergoes a spurious symmetry-breaking bifurcation. This indicates that the model is leveraging (or in this case misusing) topological constraints. Therefore, its behavior should be interpreted using the topological arguments provided by Poincaré–Hopf theorem and Proposition 4.1.
>
> We completely agree that identifying whether a model hallucinates bifurcations is vital. We propose a straightforward yet robust criterion: evaluating the variance in bifurcation diagrams generated by multiple independent training runs. Correctly identified bifurcations show minimal variance, indicating stable and consistent identification. Conversely, hallucinated bifurcations yield significantly higher variance due to their structural instability. **Figure 2** at the provided link illustrates this distinction clearly, highlighting how increased variance serves as an indicator of hallucinated behavior. We will include this result in the revised version, along with appropriate discussion.
> ***
> **Q3. Non-Hamiltonian DS.**
>
> Following your suggestion, we tested the context-informed NODEs on a non-Hamiltonian, codimension-2 cusp bifurcating system: $(\dot{x}, \dot{y}) = (y, b + a x - x^3 – d y)$, with variable parameters $(a,b)$ and fixed $d=0.5$. This system serves as a canonical model for capturing catastrophic transitions and hysteresis phenomena, which are fundamental in science and engineering. Despite its dissipative, non-Hamiltonian nature, its bounded dynamics imply a preserved total Poincaré index of +1. Training was conducted exclusively on simple, monostable pre-bifurcation conditions within the parameter range $(a,b) \in [-2.0, -1.5, -1.0, -0.5]^2$, yet the model successfully identified the cusp bifurcation surface, confirming broader applicability (**Figure 3** at the provided link). This result will be included in our revised paper.
> ***
> **Q4. Regularization without physical priors.**
>
> We agree with your comment that topological regularization requires a certain level of prior knowledge about the system. In particular, the amount of required knowledge depends on whether we apply only a global constraint or both global and local ones. Local regularization typically requires more detailed, system-specific information. In contrast, using only global constraints demands relatively less prior knowledge. As stated in the Poincaré–Hopf theorem, it depends only on coarse information about the phase manifold on which the ODE is defined. For example, in a 2D closed-orbit system, the total Poincaré index is fixed at +1. Thus, as long as we know the system is non-divergent, this global constraint can often be applied without detailed knowledge of the vector field itself. In other setting—such as when the phase manifold is a sphere—the total index is determined by the Euler characteristic. For instance, when modeling vector fields on Earth (as in climate models), it is typically reasonable to assume a global index of $\chi(S^2) = +2$.
>
> This type of global information can be especially useful when the available training data is confined to a restricted domain of initial conditions. To assess the effectiveness of topological regularization under minimal prior assumptions, we revisited the experiment in Section 5 using only the global constraint. In this additional study, we limited the training domain to $[-1.0, 1.0]^2$ to mimic a restricted coverage scenario. **Figure 5** at the provided link illustrates that the topologically regularized model exhibits improved performance, even when relying solely on global information. We will include this result in the revised paper.

---

> > ### Comment · Reviewer_XQYv · 2025-04-04
> >
> > I thank the authors for this clarifying rebuttal. I would still like to see whether the results of the paper can be applied/reproduced for general flow operator models, not just NODEs. However, I think it is fair to see this as future work. I will increase my score accordingly.

---

> > > ### Author Response · Authors · 2025-04-09
> > >
> > > We are pleased that our response has addressed your concerns. We sincerely appreciate your thoughtful suggestions and kind recognition of our work.
> > >
> > > While our current study primarily focuses on continuous dynamical systems governed by ODE vector fields, we acknowledge that incorporating alternative DSR backbones and flow models (e.g., Brenner et al. (2024)) offers an interesting opportunity to extend our framework. Such extensions could enable us to further broaden the scope of our topological perspective. We will incorporate this promising direction into our discussion of future work.

---

### Official Review · Reviewer_rQs3 · 2025-03-14

**Overall Recommendation:** 4

**Summary:**

The paper demonstrates that context-dependent Neural Ordinary Differential Equations can identify post-bifurcation behaviors, even when trained only on pre-bifurcation data. It then provides an interpretation for this phenomenon based on the Poincaré-Hopf theorem, and proposes a topological regularizer that mitigates the hallucination for post-bifurcation behaviors.

**Claims And Evidence:**

The claims seem well supported by the experiments.

**Essential References Not Discussed:**

I am not aware of any essential references not being discussed.

**Experimental Designs Or Analyses:**

The experiments are sound and well motivated.

**Methods And Evaluation Criteria:**

The evaluations make sense for the investigated problem.

**Other Comments Or Suggestions:**

1. On page 3 line 160, the left column, the definition of OOD condition for parameters in not clear for n-dimensional parameters.

**Other Strengths And Weaknesses:**

Strengths:
1. The paper is well-written and easy to follow.
2. The problem is well-motivated and the arguments and experiments are coherent.

Weaknesses:
1. As acknowledged by the authors, the scope of the investigation is limited.
2. The regularization term seems to require detailed knowledge of the dynamical system beforehand, such as the global and local contour. This may limit the applicability of the regularization.

**Questions For Authors:**

I do not have additional questions.

**Relation To Broader Scientific Literature:**

This paper studies context-dependent Neural Ordinary Differential Equations in depth on its behaviors when predicting OOD data.

**Theoretical Claims:**

I read the proof section in the appendix.

---

> ### Author Rebuttal · Authors · 2025-03-31
>
> We sincerely appreciate the reviewer’s thoughtful and constructive feedback. Below, we address each comment carefully. Additional experimental results are available in the **README** at https://github.com/anonymous-account123/icml2025-7637 and will be thoroughly incorporated into our revised paper.
> ***
> **Q1. As acknowledged by the authors, the scope of the investigation is limited.**
>
> Our analysis primarily targets closed-orbit systems (e.g., Hamiltonian flows), but the underlying principle of Proposition 4.1—the topological invariance of the total index from the Poincaré–Hopf theorem—has broader applicability. Indeed, Proposition 4.1 follows directly from Theorem 4.1 under closed orbit assumptions. Yet, the essence of the theorem lies in global topological invariants, specifically the Euler characteristic of the phase space $\mathcal{M}$ (e.g., $\chi(S^{2n}) = 2$, $\chi(S^{2n+1}) =0$,  $\chi(T^{n}) = 0$, ...), allowing its application even when the associated ODE defined on $\mathcal{M}$ does not exhibit closed orbits or conserved quantities. We will explicitly address this generalization potential in our revision.
> ***
> **Q2. The regularization term seems to require detailed knowledge of the dynamical system beforehand, such as the global and local contour. This may limit the applicability of the regularization.**
>
> We agree with your comment that topological regularization requires a certain level of prior knowledge about the system. In particular, the amount of required knowledge depends on whether we apply only a global constraint or both global and local ones. Local regularization typically requires more detailed, system-specific information. In contrast, using only global constraints demands relatively less prior knowledge. As stated in the Poincaré–Hopf theorem, it depends only on coarse information about the phase manifold on which the ODE is defined. For example, in a 2D closed-orbit system, the total Poincaré index is fixed at +1. Thus, as long as we know the system is non-divergent, this global constraint can often be applied without detailed knowledge of the vector field itself. In other setting—such as when the phase manifold is a sphere—the total index is determined by the Euler characteristic. For instance, when modeling vector fields on Earth (as in climate models), it is typically reasonable to assume a global index of $\chi(S^2) = +2$.
>
> This type of global information can be especially useful when the available training data is confined to a restricted domain of initial conditions. To assess the effectiveness of topological regularization under minimal prior assumptions, we revisited the experiment in Section 5 using only the global constraint. In this additional study, we limited the training domain to $[-1.0, 1.0]^2$ to mimic a restricted coverage scenario. **Figure 5** at the provided link illustrates that the topologically regularized model exhibits improved performance, even when relying solely on global information. We will include this result in the revised paper.
> ***
> **Q3. On page 3 line 160, the left column, the definition of OOD condition for parameters in not clear for n-dimensional parameters.**
>
> Following your suggestion, we formally describe the definition of parameter OOD conditions with codimension-$k$ bifurcation in $n$-dimensional parameter space:
>
> Let $\mathcal{P} \subset \mathbb{R}^n$ be a parameter space and let $\mathcal{B} \subset \mathcal{P}$ be a bifurcation set. The complement $\mathcal{P} \setminus \mathcal{B}$ admits a decomposition into $L$ disjoint connected components: $ \mathcal{P} \setminus \mathcal{B} = \bigcup_{i=1}^L \mathcal{P}_i$, where each $\mathcal{P}_i$ is a maximal connected subdomain. Intuitively, each $\mathcal{P}_i$ is one qualitatively uniform parameter regime. Then, a parameter OOD condition in learning dynamics arises when the support of training distribution is $\mathrm{supp}(p^\mathrm{tr}_e(\mu)) \subseteq \mathcal{P}_l$, but the support of the test distribution is $\mathrm{supp}(p^\mathrm{test}_e(\mu)) \subseteq \mathcal{P}_i$ for some $i \neq l$. Equivalently, there exists no continuous path $\gamma: [0, 1] \to \mathcal{P} \setminus \mathcal{B}$ connecting $\mu^\mathrm{tr}_e$ and $\mu^\mathrm{test}_e$ without crossing $\mathcal{B}$.
>
> This generalized notion of parameter OOD parallels the concept of initial condition OOD, enabling a unified perspective on OOD in learning dynamics. Accordingly, we will revise our paper to correctly describe this more general setting beyond the one-dimensional case.

---

> > ### Comment · Reviewer_rQs3 · 2025-04-05
> >
> > I thank the authors for addressing my comments. I do not have additional questions and will maintain my score.

---

> > > ### Author Response · Authors · 2025-04-09
> > >
> > > We are pleased that our response has addressed your concerns. We sincerely appreciate your thoughtful suggestions and kind recognition of our work.

---

### Official Review · Reviewer_RFdK · 2025-03-16

**Overall Recommendation:** 3

**Summary:**

This paper explores the use of context-informed NODEs to identify symmetry-breaking bifurcations in dynamical systems without relying on physics-based training data. The authors demonstrate that NODEs trained solely on symmetric (pre-bifurcation) data can predict post-bifurcation behaviors in a "zero-shot" manner, meaning they do so without prior experience with such data.

The key to this capability lies in the NODEs' ability to utilize topological invariants, specifically the Poincaré index, which helps them grasp the underlying structure of the data despite significant changes from bifurcations. The authors also introduce a novel topological regularizer inspired by the Poincaré-Hopf theorem to enhance NODE performance.

The paper challenges the assumption that data-driven models struggle with bifurcations and presents a promising approach that demonstrates how these models can extract meaningful dynamics from limited training data, contributing to their robustness and applicability in real-world complex systems.

**Claims And Evidence:**

Overall, their claims are almost well-supported by the evidence presented. In particular, the authors provide a solid theoretical framework grounded in the Poincaré-Hopf theorem and demonstrate the efficacy of their approach through empirical validation with the Landau-Khalatnikov system. The results show that context-informed NODEs can identify symmetry-breaking behavior without direct training on post-bifurcation data, which testifies to the power of their model. However, there are some Unclear Claims:

- The authors claim that NODEs implicitly learn topological invariants like the Poincaré index, but it is unclear if they truly "learn" them or if this just happens as a side effect of function approximation. The authors could make this claim stronger by showing exactly how the model learns these features.

- The paper presents an example in which NODEs "hallucinate" broken symmetry when trained on a linear system (Sec. 4.3). But, it is unclear whether this phenomenon is a general characteristic of NODEs or if it is specific to the chosen experimental setup.

There are also some concerns that could impact the robustness of their claims more generally in more complex scenarios:

- The study predominantly investigates **codimension-one** bifurcations, focusing solely on **2D** and **Hamiltonian** systems, which narrows the applicability of their findings. Although Theorem 4.1 hints at potential extensions to more complex systems or higher dimensions, it would strengthen the argument if the authors elaborated on potential strategies for tackling **codimension-higher bifurcations**, the challenges they might face in 3D or higher-dimensional systems, and the adaptations required in their approach for non-Hamiltonian systems. It would be helpful to provide insights into how these generalizations might occur and under what specific conditions it would hold true.

**Essential References Not Discussed:**

-

**Experimental Designs Or Analyses:**

Re soundness/validity of experimental designs or analyses:

**Topological Regularization**: They introduce a Poincaré-Hopf regularization aimed at enhancing the NODEs' learning capabilities. However, it lacks detailed information on how this regularization is specifically implemented and tuned, which is essential for assessing its effectiveness and reproducibility.

**Model Evaluation**: The evaluation metric relies on checking if the model predictions converge within a distance (σ) from the true dynamics. If this distance threshold is not well-defined or justified, it could lead to misleading conclusions about the model's predictive capabilities and overall performance.

**Parameter Exploration**:  A mesh grid approach is used for parameter sampling, but the methodology for how this grid is constructed and the resolution of the sampling are crucial.

**Methods And Evaluation Criteria:**

**Proposed Methods**: First off, they have nicely discussed the OOD generalization problem by categorizing it into two aspects: OOD in initial conditions and OOD in model parameters, to better clarify the two different challenges for learning dynamics. The use of context-informed NODEs and Poincaré-Hopf regularization makes sense for studying bifurcations and symmetry-breaking in dynamical systems.

**Evaluation Approach**: The paper evaluates NODEs on well-defined 2D dynamical systems (e.g., Hamiltonian and Landau-Khalatnikov models), which are relevant for studying bifurcations. The experiments mostly involve Hamiltonian or nearly conservative systems and focus only on codimension-one bifurcations. Yet, real-world systems often include dissipation and forcing, which can alter bifurcation behavior and lead to codimension-higher bifurcations. So, evaluating on non-Hamiltonian, higher-dimensional, and multi-parameter bifurcations ($ \mu \in \mathbb{R}^n, n>1 $) would provide a more comprehensive validation.

**Other Comments Or Suggestions:**

**Some minor comments**:

- Page 1, second column, line 21: It is mentioned "Formally, a dynamical system is represented by a phase space Ordinary Differential Equation (ODE)". But, it is better to say "Formally, many **continuous-time** dynamical system ... " as discrete-time dynamical systems can be represented by recursive maps. Also, some continuous-time dynamical systems can be represented by PDEs, not necessarily ODEs.

- Page 3, in Sect. "OOD in model parameters":  The definition of bifurcation parameter (starting from line 151) is for the general case $\mu_{crit} \in \mathbb{R}^n$. But, the sentence "Consequently, the OOD condition for parameters arises when a model is trained on parameters $\mu_e^{tr} < \mu_{crit}$ but the support of the test data is $\mu_e^{test} > \mu_{crit}$"  is only valid for $\mu_e^{tr} ,  \mu_{crit} ,  \mu_{crit} \in \mathbb{R}$ (codimension-one bifurcation). Please clarify it in the text.

- Page 6, Definition 4.1: The topological degree of a map should be better clarified/explained.

**Other Strengths And Weaknesses:**

**Other Strengths**:

- Originality:  The paper introduces a fresh approach by showing how context-dependent NODEs, trained on localized data, can effectively identify symmetry-breaking bifurcations. This highlights a creative use of topological invariants in a data-driven setting.

- Significance: By tackling how NODEs can extrapolate behaviors beyond their training domain, this research makes a meaningful contribution to both dynamical systems and machine learning. It could open up new possibilities for automating scientific discovery and deepens our understanding of complex systems.

- Clarity: The paper is almost well-written and well-structured and explains complex ideas in a way that's easy to follow.

**Questions For Authors:**

1. In Section 3.1, could you elaborate on the rationale for randomly sampling **four** initial conditions? Furthermore, what is the justification for adapting the model using a **single** trajectory after training it with pre-bifurcation data?

2. In Section 2, you introduce a Poincaré-Hopf regularization to enhance the NODEs' learning capabilities. Coulld you provide more detailed information on the specific implementation and tuning process of this regularization? How do you ensure that it effectively contributes to the learning without introducing unwanted biases?

3. In Section 5, the evaluation of model performance is based on the convergence within a distance (σ) from the true dynamics. How is this distance threshold (σ) determined, and what justification or heuristics do you provide for selecting its value? Could different choices of σ significantly affect your conclusions regarding the model's predictive capabilities?

4. Could you clarify how far your results, particularly Proposition 4.1, can be generalized to non-Hamiltonian systems? What specific characteristics or conditions of non-Hamiltonian systems would need to be considered for the applications of Proposition 4.1 to remain valid, and how might the presence of dissipation or external forcing impact the conclusions drawn from your study regarding symmetry-breaking behavior?

5. The authors claim that NODEs implicitly learn topological invariants like the Poincaré index, but it is unclear if they truly "learn" them or if this just happens as a side effect of function approximation. Could elaborate on this further and make this claim stronger by showing exactly how the model learns these features?

I am happy to increase my score if the authors can address my main concerns.

**Relation To Broader Scientific Literature:**

The key contributions of the paper relate to several areas, particularly regarding modeling dynamical systems and understanding bifurcations through data-driven approaches. It has implications for theoretical advancements in dynamical systems and OOD generalization challenges. Specifically, it discusses OOD in model parameters, which is another category of OOD generalization challenges, in addition to OOD in initial conditions as noted in  (G¨oring et al., 2024), which is very important.

**Key Points**:

- Data-Driven Dynamics: Builds on Brunton et al. (2016), demonstrating effective discovery of dynamical systems without physics-based priors.

- Topological Invariants: Utilizes invariants (Brasselet et al., 2009) to enhance NODEs by linking topology and dynamics.

- OOD Generalization: Challenges beliefs about model generalization across bifurcations (Ye et al., 2021), showing robust NODE performance without diverse training.

- Symmetry Recovery: Contrasts with García Pérez et al. (2023), indicating NODEs' ability to model symmetry-breaking without physics-informed priors.

**Theoretical Claims:**

I tried to verify the correctness of the theoretical claims in the main text as well as the proofs in Appendix A.

**Re the proof of Proposition 4.1**:
- The proposition outlines a condition for local symmetry breaking in a Hamiltonian vector field via a center-to-saddle bifurcation. This condition requires a smooth bijective map that maintains vector fields close in the $C^1$ sense, potentially overlooking complexities in real-world systems where perturbations are not small/easily managed.

- The proof logically follows from bifurcation theory, Implicit Function Theorem and $C^1$ closeness assumptions between the learned and true **Hamiltonian** vector fields. The proof assumes the learned vector field closely matches the true system, but in practice, Neural ODEs are approximations and may not perfectly capture the system's dynamics. If there are errors in approximation (due to limited training data, network capacity, or optimization issues), the conditions stated in the proof might not always hold, making the model’s predicted bifurcations less reliable in real-world applications.

- The proof in Proposition 4.1 is specifically derived for Hamiltonian systems, where energy conservation and structured dynamics naturally enforce certain constraints. However, many real-world systems are non-Hamiltonian which can significantly alter bifurcation behavior. Since the proof relies on preserving closed orbits and the Poincaré-Hopf index, its conclusions may not directly apply to non-Hamiltonian systems where these properties don't hold. The authors should clarify how far their results, particularly Proposition 4.1, can be generalized to non-Hamiltonian systems. This includes discussing the limitations of their findings and what further theoretical or empirical validation is needed to adapt their conclusions for these systems.

---

> ### Author Rebuttal · Authors · 2025-03-31
>
> We sincerely appreciate the reviewer’s thoughtful and constructive feedback. Below, we address each comment carefully. Additional experimental results are available in the **README** at https://github.com/anonymous-account123/icml2025-7637 and will be thoroughly incorporated into our revised paper.
> ***
> **Q1. Do NODEs genuinely utilize topology? Is the hallucinated broken symmetry general?**
>
> You highlight a critical point about distinguishing whether context-informed NODEs genuinely leverage topological constraints or merely approximate functions that happen to exhibit topological properties. Indeed, the double-well scenario poses a dilemma: better approximations naturally reflect correct symmetry-breaking behavior. Our hallucinated bifurcation scenario in Section 4.3, grounded explicitly in Proposition 4.1, is intentionally designed to address this challenge. If NODEs were merely approximating a functional form, a simple center-to-saddle bifurcation would emerge. Instead, the model exhibits an incorrect symmetry-breaking bifurcation, confirming the model genuinely utilizes (in this scenario, misuses) topological constraints, precisely as Proposition 4.1 predicts. Hence, our findings demonstrate that NODEs indeed exploit topological invariants. In addition, the hallucinated symmetry-breaking can arise whenever Proposition 4.1's conditions hold. We will highlight this more clearly in the revised manuscript.
> ***
> **Q2. Robustness of bifurcation identification under noisy conditions.**
>
> We acknowledge that approximation errors may increase under noisy perturbations or with limited training samples, as you pointed out. To address this, we conducted further experiments under these realistic conditions. Despite the increased variance, the model consistently identified the symmetry-breaking bifurcation, demonstrating its robustness. The results are illustrated in **Figure 1(b-c)** at the provided link, which will be included in the revised manuscript.
> ***
> **Q3. Non-Hamiltonian and codimension-higher bifurcations.**
>
> Following your suggestion, we tested the context-informed NODEs on a non-Hamiltonian, codimension-2 cusp bifurcating system: $(\dot{x}, \dot{y}) = (y, b + a x - x^3 – d y)$, with variable parameters $(a,b)$ and fixed $d=0.5$. This system serves as a canonical model for capturing catastrophic transitions and hysteresis phenomena. Despite its dissipative, non-Hamiltonian nature, its bounded dynamics imply a preserved total Poincaré index of +1. Training was conducted exclusively on simple, monostable pre-bifurcation conditions within the parameter range $(a,b) \in [-2.0, -1.5, -1.0, -0.5]^2$, yet the model successfully identified the cusp bifurcation surface, confirming broader applicability (**Figure 3** at the provided link). This result will be included in our revised paper.
> ***
> **Q4. Generalization of Proposition 4.1.**
>
> Our analysis primarily targets closed-orbit systems (e.g., Hamiltonian flows), but the underlying principle of Proposition 4.1—the topological invariance of the total index from the Poincaré–Hopf theorem—has broader applicability. Indeed, Proposition 4.1 follows directly from Theorem 4.1 under closed orbit assumptions. Yet, the essence of the theorem lies in global topological invariants, specifically the Euler characteristic of the phase space $\mathcal{M}$ (e.g., $\chi(S^{2n}) = 2$, $\chi(S^{2n+1}) = 0$, $\chi(T^{n}) = 0$, ...), allowing its application even when the associated ODE defined on $\mathcal{M}$ does not exhibit closed orbits or conserved quantities. We will explicitly address this generalization potential in our revision.
> ***
> **Q5. Experimental designs.**
>
> We have provided a detailed description and implementation of our proposed topological regularization approach in Appendix Section G. Additionally, Figure 17 in Appendix illustrates consistency scores across various threshold values ($\sigma$) ranging from 0.2 to 1.0, for the results of Section 5. The regularized model consistently outperforms vanilla models, maintaining scores close to 1.0 regardless of $\sigma$. Furthermore, Appendix Section H thoroughly summarizes our experimental setup (e.g., 16 training and 441 testing parameter combinations over an $(\alpha,\beta)$ grid).
> ***
> **Q6. Why four training samples and one for adaptation?**
>
> Our experimental setup closely mirrors Kirchmeyer et al. (2022), who utilized four initial conditions per parameter for training on the Lotka–Volterra system—another 2D closed-orbit system analogous to ours. For adaptation, a single confined trajectory was selected intentionally to simulate an extreme symmetry-breaking scenario. This deliberately restricts the model’s exposure to the global phase space structure, effectively testing simultaneous parameter and initial condition OOD challenges.
> ***
> **Q7. Minor comments.**
>
> We will clarify the manuscript as recommended. For the clarified definition of OOD in parameters, please refer to our response to Question 3 from Reviewer rQs3.

---

### Decision · Program_Chairs · 2025-05-01

**Decision:**

Accept (poster)

**Comment:**

After the discussion, the reviews are all positive. The paper proposes a method for NODE network to learn symmetries in dynamical systems as supported by a complete theoretical analysis. It clearly reaches the level of quality and novelty for ICML.